# ENHANCED DIFFUSION SAMPLING VIA EXTRAPOLATION WITH MULTIPLE ODE SOLUTIONS

**Jinyoung Choi[1], Junoh Kang[1] & Bohyung Han[1,2]**
Computer Vision Lab., [1]ECE & [2]IPAI, Seoul National University, Korea
{jin0.choi, junoh.kang, bhhan}@snu.ac.kr

## ABSTRACT

Diffusion probabilistic models (DPMs), while effective in generating high-quality samples, often suffer from high computational costs due to their iterative sampling process. To address this, we propose an enhanced ODE-based sampling method for DPMs inspired by Richardson extrapolation, which reduces numerical error and improves convergence rates. Our method, RX-DPM, leverages multiple ODE solutions at intermediate time steps to extrapolate the denoised prediction in DPMs. This significantly enhances the accuracy of estimations for the final sample while maintaining the number of function evaluations (NFEs). Unlike standard Richardson extrapolation, which assumes uniform discretization of the time grid, we develop a more general formulation tailored to arbitrary time step scheduling, guided by local truncation error derived from a baseline sampling method. The simplicity of our approach facilitates accurate estimation of numerical solutions without significant computational overhead, and allows for seamless and convenient integration into various DPMs and solvers. Additionally, RX-DPM provides explicit error estimates, effectively demonstrating the faster convergence as the leading error term's order increases. Through a series of experiments, we show that the proposed method improves the quality of generated samples without requiring additional sampling iterations.

## 1 INTRODUCTION

Diffusion probabilistic models (DPMs) have emerged as a powerful framework for generating high-quality samples in a wide range of applications and domains for images (Ho et al., 2020; Song et al., 2021b; Dhariwal & Nichol, 2021; Rombach et al., 2022), videos (Ho et al., 2022; Singer et al., 2022; Zhou et al., 2022; Wang et al., 2023), 3D shapes (Zeng et al., 2022), *etc*. While DPMs demonstrate impressive performance in data fidelity and diversity, they also have limitations, particularly their computational inefficiency due to the sequential nature of sampling. Addressing this issue is crucial for enhancing the usability of DPMs in real-world scenarios, where time constraints are critical for practical deployment.

The generation process of DPMs can be formulated as a problem of finding solutions to SDEs or ODEs (Song et al., 2021b), where the truncation errors of the numerical solutions are highly correlated to the quality of the generated samples. To enhance the quality of these samples, it is essential to reduce truncation errors, which can be achieved by adopting advanced solvers or numerical techniques that improve the accuracy of numerical estimations. In this context, we aim to lower truncation errors by applying numerical extrapolation to existing sampling methods for DPMs. The key ingredient of the proposed method is Richardson extrapolation, a proven and widely used technique in the mathematical modeling of physical problems such as fluid dynamics and heat transfer, which demand high computational resources. While numerous variants and strategies have been studied (Richards, 1997; Botchev & Verwer, 2009; Zlatev et al., 2010), its application to DPMs remains unexplored. The method uses a simple linear combination of multiple numerical estimates from different resolutions of a grid to approximate the ideal solution, where the estimates are expected to converge as the resolution becomes finer, ultimately reaching the target value in the limit.

We propose an extrapolation algorithm that is applied repeatedly every $k$ denoising steps of an ODE-based sampling method to improve the accuracy of intermediate denoising steps. This is achieved by utilizing an additional ODE solution, which is estimated by a single step over an interval of $k$ time steps. Figure 1 illustrates this concept with $k = 2$ on time steps $[t_i, t_{i-1}, t_{i-2}]$, which forms a unit block of our extrapolation-based sampling. Two ODE solutions—single-step and two-step estimations at $t_{i-2}$ from $t_i$—can be leveraged to achieve an approximation closer to the ideal solution $\boldsymbol{x}^*_{t_{i-2}}$, which is unknown.

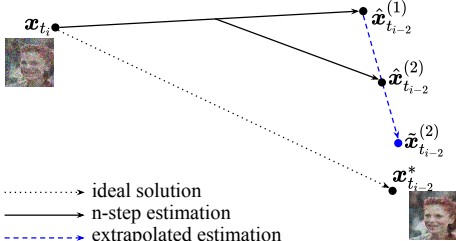

Figure 1: Application of the proposed extrapolation on two denoising steps ($k = 2$) with time steps of $[t_i, t_{i-1}, t_{i-2}]$. $\hat{\boldsymbol{x}}^{(n)}_{t_{i-2}}$ denotes that $n$ steps are used by the baseline sampler within the same interval. $\tilde{\boldsymbol{x}}^{(2)}_{t_{i-2}}$ represents the extrapolated estimation using two ODE solutions at $t_{i-2}$, $\hat{\boldsymbol{x}}^{(1)}_{t_{i-2}}$ and $\hat{\boldsymbol{x}}^{(2)}_{t_{i-2}}$.

The standard Richardson extrapolation assumes a uniform discretization of the time grid. However, the uniform grid of denoising time steps might be suboptimal for DPMs; a smaller time interval near the clean sample is often more beneficial (Karras et al., 2022; Song et al., 2021a) given the same number of steps. Considering such characteristics of DPMs and the advantages offered by specific time step discretizations, we propose an algorithm applicable to arbitrary schedules, with coefficients determined by the chosen configuration. We observe that our grid-aware approach yields better performance than conventional methods.

Although there exist other methods applying extrapolation techniques to diffusion models, their usages of extrapolation are somewhat different from ours. For example, Zhang & Chen (2023); Zhang et al. (2023) utilize estimations from earlier steps to improve the estimation at the time step, $t_i$, whereas our approach adopts two denoised estimations at the same time step, $t_i$, to enhance accuracy at $t_i$. In addition, the main building block of our approach, Richardson extrapolation, is proven to enhance numerical accuracy and provides an explicit estimate of the error, which allows for a clear understanding of the convergence behavior. Furthermore, the implementation of our algorithm is simple and cost-effective because it requires no additional network evaluations and insignificant computational overhead to perform the extrapolation. We refer to the proposed sampling algorithm as RX-DPM.

Our main contributions are summarized below:

- We introduce an improved diffusion sampler, RX-DPM, inspired by Richardson extrapolation, which effectively increases the order of accuracy of existing ODE-based samplers without increasing NFEs.
- We systematically develop an algorithm for general DPM solvers with arbitrary time step scheduling starting from the derivation of a truncation error of the Euler method on a non-uniform grid.
- Our experiments across various well-known baselines demonstrate that RX-DPM exhibits strong generalization performance and high practicality, regardless of ODE designs, model architectures, and base samplers.

## 2 RELATED WORK

There exists a substantial body of research that seeks to reduce the computational burden of DPMs while maintaining their performance. One approach in this direction involves exploring alternative modeling strategies for the reverse process. For example, the networks in Salimans & Ho (2022); Song et al. (2023); Kim et al. (2024) learn alternative objectives through knowledge distillation, using the outputs obtained by iterative inferences of the pretrained (teacher) networks. On the other hand, Bao et al. (2022) model a more accurate reverse distribution by incorporating the optimal covariance, while Xiao et al. (2022); Kang et al. (2024) implicitly estimate the precise reverse distribution by utilizing GAN components.

While the aforementioned methods require model training, there is also a training-free approach that interprets the generation process of a diffusion model as solving an ODE or SDE (Song et al., 2021b;

Karras et al., 2022). For instance, DDIM (Song et al., 2021a) proposes to skip intermediate time steps, which is equivalent to solving an ODE using the Euler method with a large step size. To further improve sampling quality, a large volume of research (Karras et al., 2022; Dockhorn et al., 2022; Liu et al., 2022; Zhang & Chen, 2023; Lu et al., 2022; 2023) simply applies classical higher-order solvers or customizes them for diffusion models. Specifically, Karras et al. (2022) adopt the second-order Heun's method (Süli & Mayers, 2003) and Dockhorn et al. (2022) apply the second-order Taylor expansion. In addtion, Liu et al. (2022) propose a pseudo-numerical solver, which combines classical high-order numerical methods and DDIM.

Building on ODE-based sampling methods, various numerical techniques have been employed to further improve the sample quality. For example, IIA (Zhang et al., 2024) optimizes the coefficients of certain quantities to approximate the fine-grained integration. LA-DPM (Zhang et al., 2023) and DEIS (Zhang & Chen, 2023) adopt extrapolation techniques as our method, but with notable differences. Specifically, LA-DPM linearly extrapolates the previous and current predictions of the solution at $t = 0$—the original data manifold—while DEIS uses high-order polynomial extrapolation on the noise prediction function. The key distinction of our approach is that, while they adopt score (noise) predictions obtained at different time steps for each extrapolation, we utilize multiple denoised outputs obtained at the same time step.

## 3 PRELIMINARIES

### 3.1 DIFFUSION PROBABILISTIC MODELS AS SOLVING AN ODE

For $\boldsymbol{p}_0 = \boldsymbol{p}_{\text{data}}$ and $\boldsymbol{x} \in \mathbb{R}^d$, Karras et al. (2022) defines a marginal distribution at $t$ as

$$\boldsymbol{p}_t(\boldsymbol{x}) = s(t)^{-d}\boldsymbol{p}(\boldsymbol{x}/s(t); \sigma(t)), \tag{1}$$

where $\boldsymbol{p}(\boldsymbol{x}; \sigma) = \boldsymbol{p}_{\text{data}} * \mathcal{N}(\boldsymbol{0}, \sigma(t)^2 \boldsymbol{I})$, and $s(t)$ and $\sigma(t)$ are non-negative functions satisfying $s(0) = 1$, $\sigma(0) = 0$, and $\lim_{t\to\infty} \frac{\sigma(t)}{s(t)} = \infty$. The probability flow ODE,

$$d\boldsymbol{x} = [\dot{s}(t)/s(t) - s(t)^2\dot{\sigma}(t)\sigma(t)\nabla_{\boldsymbol{x}} \log p(\boldsymbol{x}/s(t); \sigma(t))]dt, \quad \boldsymbol{x}(T) \sim \boldsymbol{p}_T(\boldsymbol{x}), \tag{2}$$

matches the marginal distribution. By adopting the specific choices, $s(t) = 1$ and $\sigma(t) = t$ as in Karras et al. (2022), Equation (2) is reduced as follows:

$$d\boldsymbol{x} = -t\nabla_{\boldsymbol{x}} \log \boldsymbol{p}(\boldsymbol{x}; t)dt, \quad \boldsymbol{x}(T) \sim \boldsymbol{p}_T(\boldsymbol{x}). \tag{3}$$

Diffusion models now learn the score function $\nabla_{\boldsymbol{x}} \log \boldsymbol{p}(\boldsymbol{x}; t)$, which is the only unknown component in the equation. For sufficiently large $T$, the marginal distribution $\boldsymbol{p}_T(\boldsymbol{x})$ can be approximated by $\mathcal{N}(\boldsymbol{x}; \boldsymbol{0}, T^2\boldsymbol{I})$ and the generation process is equivalent to solving for $\boldsymbol{x}(0)$ using Equation (3) with the boundary condition, $\boldsymbol{x}(T) \sim \mathcal{N}(\boldsymbol{0}, T^2\boldsymbol{I})$. Since the analytic solution of Equation (3) cannot be expressed in a closed form, numerical methods are used to solve the ODE. Given the time step scheduling, $0 = t_0 < t_1 < \ldots < t_N = T$, the solution is given by

$$\boldsymbol{x}(0) = \boldsymbol{x}(T) + \int_T^0 -t\nabla_{\boldsymbol{x}} \log \boldsymbol{p}(\boldsymbol{x}(t); t)dt \tag{4}$$

$$= \boldsymbol{x}(T) + \sum_{i=N}^1 \int_{t_i}^{t_{i-1}} -t\nabla_{\boldsymbol{x}} \log \boldsymbol{p}(\boldsymbol{x}(t); t)dt, \tag{5}$$

where each integration from $t_i$ to $t_{i-1}$ can be approximated by ODE solvers such as the Euler method or Heun's method.

### 3.2 RICHARDSON EXTRAPOLATION

Let the exact and numerical solutions at $t = 0$ be $V^*$ and $V(h)$, respectively, where $h$ ($0 < h < 1$) denotes the step size. If $V^* = \lim_{h\to 0} V(h)$ and the order of truncation error is known, Richardson extrapolation (Richardson, 1911) identifies a faster converging sequence, $\tilde{V}(h)$. For instance, $V(h)$ with a truncation error in the order of $O(h^p)$ is expressed by

$$V^* = V(h) + ch^p + O(h^q) \tag{6}$$

for $0 < p < q$ and $c \neq 0$. Then, for a fixed constant $k > 1$,

$$V^* = V(h/k) + \frac{c}{k^p}h^p + O(h^q). \tag{7}$$

From Equations (6) and (7), eliminating the $h^p$ terms, we obtain the solution

$$\tilde{V}(h,k) = \frac{k^p V(h/k) - V(h)}{k^p - 1}, \tag{8}$$

which has a truncation error of $O(h^q)$, asymptotically smaller than $O(h^p)$.

## 4   RX-DPM

Before discussing the proposed method, we first outline the algorithmic development process for the most simplified problem and then explore an extension to a general DPM solver. The application of our method, RX-DPM, to a specific solver will be referred to as RX-[*SolverName*].

### 4.1   TRUNCATION ERROR OF EULER METHOD ON NON-UNIFORM GRID

We now derive the truncation error formula for the Euler method on a non-uniform grid, based on the local truncation error, which results from a single iteration. For intuitive clarity, we consider a one-dimensional ODE of the form

$$dx = f(x, t)dt,$$

where $f$ is a smooth function. Suppose that the numerical solution is obtained using the Euler method with the discretization points $[t_i, t_{i-1}, \ldots, t_{i-k}]$ in a reverse time order, given the initial condition $x(t_i) = x_{t_i}$. From now on, we denote $\hat{x}_{t_j}^{(n)}$ as the numerical solution at $t_j$ obtained by $n$ iterations and $x_{t_j}^*$ as the exact solution at $t_j$. Given $h = t_i - t_{i-k}$ and $\lambda_j = \frac{1}{h}(t_{i-j+1} - t_{i-j})$ for $j = 1, \ldots, k$, the local truncation error formula of the one-step Euler method, derived from the Taylor expansion, is expressed as

$$\hat{x}_{t_{i-1}}^{(1)} = x_{t_i} - \lambda_1 h f(x_{t_i}; t_i) = x_{t_{i-1}}^* - \frac{1}{2}x_{t_i}'' \lambda_1^2 h^2 + O(h^3). \tag{9}$$

Then, the truncation error of the two-step numerical solution is derived as

$$\hat{x}_{t_{i-2}}^{(2)} = \hat{x}_{t_{i-1}}^{(1)} - \lambda_2 h f(\hat{x}_{t_{i-1}}^{(1)}) \tag{10}$$

$$= x_{t_{i-1}}^* - \frac{1}{2}x_{t_i}'' \lambda_1^2 h^2 + O(h^3) - \lambda_2 h f(\hat{x}_{t_{i-1}}^{(1)}) \tag{11}$$

$$= x_{t_{i-1}}^* - \lambda_2 h f(x_{t_{i-1}}^*) - \frac{1}{2}x_{t_i}'' \lambda_1^2 h^2 + O(h^3) - \lambda_2 h f(\hat{x}_{t_{i-1}}^{(1)}) + \lambda_2 h f(x_{t_{i-1}}^*) \tag{12}$$

$$= x_{t_{i-2}}^* - \frac{1}{2}x_{t_{i-1}}^{*''} \lambda_2^2 h^2 - \frac{1}{2}x_{t_i}'' \lambda_1^2 h^2 + O(h^3) \quad (\because \text{Equation (9) and } f \text{ is smooth}) \tag{13}$$

$$= x_{t_{i-2}}^* - \frac{1}{2}x_{t_i}'' (\lambda_1^2 + \lambda_2^2) h^2 + O(h^3) \quad (\because f \text{ is smooth}). \tag{14}$$

Inductively, we can obtain the truncation error for the $k$-step solution as

$$\hat{x}_{t_{i-k}}^{(k)} = x_{t_{i-k}}^* - \frac{1}{2}x_{t_i}'' \sum_{j=1}^{k} \lambda_j^2 h^2 + O(h^3), \tag{15}$$

which approximates $x_{t_{i-k}}^*$ with a truncation error of $O(h^2)$.

### 4.2   RX-EULER

We now describe RX-Euler performing extrapolation every $k$ steps on the Euler method. Extrapolation is executed as a linear combination of two different numerical solutions $\hat{x}_{t_{i-k}}^{(1)}$ and $\hat{x}_{t_{i-k}}^{(k)}$ obtained by the Euler solver over a single step on the grid $[t_i, t_{i-k}]$ and $k$ steps on the grid $[t_i, t_{i-1}, \ldots, t_{i-k}]$,

respectively. To calculate coefficients for extrapolation, we use the truncation error derived in Section 4.1, which can be also applied to Equation (3) in Section 3.1, as the ideal score function is considered smooth; its derivative is Lipschitz continuous, referring to the equation in Appendix B.3 of Karras et al. (2022). From Equations (9) and (15), we derive the expressions for $\hat{\boldsymbol{x}}_{t_{i-k}}^{(1)}$ and $\hat{\boldsymbol{x}}_{t_{i-k}}^{(k)}$ for a constant $\boldsymbol{c}$ as follows:

$$\hat{\boldsymbol{x}}_{t_{i-k}}^{(1)} = \boldsymbol{x}_{t_{i-k}}^* + \boldsymbol{c}h^2 + O(h^3) \tag{16}$$

$$\hat{\boldsymbol{x}}_{t_{i-k}}^{(k)} = \boldsymbol{x}_{t_{i-k}}^* + \boldsymbol{c}\sum_{j=1}^{k} \lambda_j^2 h^2 + O(h^3). \tag{17}$$

Then, by solving the linear system of Equations (16) and (17), we approximate $\boldsymbol{x}_{t_{i-k}}^*$ through the following extrapolation:

$$\tilde{\boldsymbol{x}}_{t_{i-k}}^{(k)} = \frac{\hat{\boldsymbol{x}}_{t_{i-k}}^{(k)} - \sum_{j=1}^{k} \lambda_j^2 \hat{\boldsymbol{x}}_{t_{i-k}}^{(1)}}{1 - \sum_{j=1}^{k} \lambda_j^2}, \tag{18}$$

which involves a truncation error of $O(h^3)$, asymptotically smaller than $O(h^2)$.

In the sampling process, we set the initial condition at the next denoising step, $t_{i-k}$, as $\tilde{\boldsymbol{x}}_{t_{i-1}}^{(k)}$, and repeatedly perform the proposed extrapolation technique every $k$ steps. Because this approach provides provably more accurate solutions at every $k$ steps, we can reduce error propagation and expect better quality of generated examples.

The proposed method is applicable to first-order methods in general, including DDIM (Song et al., 2021a), which is arguably the most widely used DPM sampler. In this context, we bring the interpretation of DDIM as the Euler method applied to the following ODE:

$$d\boldsymbol{y} = \epsilon_\theta(\boldsymbol{x}(t), t)d\gamma, \tag{19}$$

where $\boldsymbol{y}(t) = \boldsymbol{x}(t)\sqrt{1 + \gamma(t)^2}$ and $\gamma(t) = \sqrt{\frac{1 - \alpha_t^2}{\alpha_t^2}}$ in the variance-preserving diffusion process (Song et al., 2021b), *i.e.*, $\boldsymbol{p}_t(\boldsymbol{x}|\boldsymbol{x}_0) = \mathcal{N}(\alpha_t \boldsymbol{x}_0, (1 - \alpha_t^2)\boldsymbol{I})$. Thus, instead of using the time grid, we compute $\lambda_j$'s in Equation (18) in terms of $\gamma(t)$'s, replacing $t$ with the corresponding $\gamma(t)$ in the computation, while the other procedures remain unchanged.

RX-Euler (RX-DDIM) does not require additional NFEs beyond the number of time steps, as the first prediction of every $k$-step-interval can be stored during the computation of $\hat{\boldsymbol{x}}^{(k)}$ and reused to obtain $\hat{\boldsymbol{x}}^{(1)}$. The only extra computation involves a linear combination of two estimates, which is negligible compared to the forward evaluations of DPMs.

## 4.3 RX-DPM WITH HIGHER-ORDER SOLVERS

We now present the algorithm for general ODE samplers of DPMs including high-order solvers. When the extrapolation occurs every $k$ steps, as in Section 4.2, the error of $\hat{\boldsymbol{x}}_{t_{i-k}}^{(1)}$, resulting from a single iteration of an arbitrary ODE method, is given by

$$\hat{\boldsymbol{x}}_{t_{i-k}}^{(1)} = \boldsymbol{x}_{t_{i-k}}^* + \boldsymbol{c}h^p + O(h^q) \tag{20}$$

for $0 < p < q$ and $\boldsymbol{c} \neq \boldsymbol{0}$. For $\hat{\boldsymbol{x}}_{t_{i-k}}^{(k)}$, we extend the form of the linear error accumulation observed in Equation (17) to obtain the following equation:

$$\hat{\boldsymbol{x}}_{t_{i-k}}^{(k)} = \boldsymbol{x}_{t_{i-k}}^* + \boldsymbol{c}\sum_{j=1}^{k} \lambda_j^p h^p + O(h^q). \tag{21}$$

Note that Equation (21) does not hold in general; however, we consider this simplified assumption reasonable, as it is consistent with the standard assumption of Richardson extrapolation under uniform discretization (see Appendix A). Finally, by solving the linear system of Equations (20) and (21), the extrapolated solution is given by

$$\tilde{\boldsymbol{x}}_{t_{i-k}}^{(k)} = \frac{\hat{\boldsymbol{x}}_{t_{i-k}}^{(k)} - \sum_{j=1}^{k} \lambda_j^p \hat{\boldsymbol{x}}_{t_{i-k}}^{(1)}}{1 - \sum_{j=1}^{k} \lambda_j^p}, \tag{22}$$

which approximates $\boldsymbol{x}^*_{t_{i-k}}$ with a truncation error of $O(h^q)$, asymptotically smaller than $O(h^p)$.

A limitation of this approach is that estimating $\hat{\boldsymbol{x}}^{(1)}_{t_{i-k}}$ and $\hat{\boldsymbol{x}}^{(k)}_{t_{i-k}}$ through naïve applications of higher-order solvers requires additional network evaluations compared to baseline sampling. However, since $\hat{\boldsymbol{x}}^{(1)}_{t_{i-k}}$ is accessible from the network predictions made for the computation of $\hat{\boldsymbol{x}}^{(k)}_{t_{i-k}}$, applying our method to higher-order solvers does not increase the NFE provided that intermediate predictions are properly stored. We next discuss how this is achieved using specific examples of high-order ODE solvers; the generalization to other solvers is mostly straightforward.

Before moving forward, we note that high-order solvers typically rely on interpolation-based techniques, such as the Runge-Kutta method (Süli & Mayers, 2003) and linear multistep method (Timothy, 2017), where the former employs evaluations at multiple intermediate points, while the latter leverages evaluations from previous steps.

**RX-Runge-Kutta**   We consider applying our method with $k = 2$ to the second-order Runge-Kutta method. A sequence of one-step estimates are given by

$$\hat{\boldsymbol{x}}^{(1)}_{t_{i-1}} = \boldsymbol{x}_{t_i} - (t_i - t_{i-1})(a_1 \mathbf{z}_i + a_2 \mathbf{z}_{i-\delta}) \text{ and} \tag{23}$$

$$\hat{\boldsymbol{x}}^{(2)}_{t_{i-2}} = \hat{\boldsymbol{x}}^{(1)}_{t_{i-1}} - (t_{i-1} - t_{i-2})(a_1 \mathbf{z}_{i-1} + a_2 \mathbf{z}_{i-1-\delta}). \tag{24}$$

where $\mathbf{z}_j = \epsilon_\theta(\boldsymbol{x}(t_j), t_j)$ for $t_{j-1} < t_{j-\delta} \leq t_j$. Then, we can express the single combined-step estimate at $t_{i-2}$ as

$$\hat{\boldsymbol{x}}^{(1)}_{t_{i-2}} = \boldsymbol{x}_{t_i} - (t_i - t_{i-2})(a_1 \mathbf{z}_i + a_2 \mathbf{z}_{i-\delta'}). \tag{25}$$

Since $\mathbf{z}_i$ is reusable after the calculation of $\hat{\boldsymbol{x}}^{(1)}_{t_{i-1}}$, we only need to compute $\mathbf{z}_{i-\delta'}$, which is approximated as $\mathbf{z}_{i-1}$ or $\mathbf{z}_{i-1-\delta}$, depending on the proximity of its time step. This approach allows us to efficiently extrapolate the solutions without compromising the quality of the generated samples.

**RX-Adams-Bashforth**   Suppose that, by the $s$-step Adams-Bashforth method, extrapolation is performed on a grid with an interval of $h$ every $k$ steps. For predefined $b_j$'s, we are given

$$\hat{\boldsymbol{x}}^{(k)}_{t_{i-k}} = \hat{\boldsymbol{x}}_{t_{i-k+1}} + h \sum_{j=0}^{s} b_j \epsilon_\theta(\hat{\boldsymbol{x}}_{t_{i-k+j}}, t_{i-k+j}). \tag{26}$$

Then, we compute $\hat{\boldsymbol{x}}^{(1)}_{t_{i-k}}$ with a step size of $kh$ for extrapolation as

$$\hat{\boldsymbol{x}}^{(1)}_{t_{i-k}} = \hat{\boldsymbol{x}}_{t_i} + kh \sum_{j=0}^{s} b_j \epsilon_\theta(\hat{\boldsymbol{x}}_{t_{i-k+jk}}, t_{i-k+jk}) \tag{27}$$

which requires no additional NFEs by storing the previous network evaluations.

Algorithm 1 summarizes the procedure of the proposed method with a generic ODE solver under the assumption that $N$ is a multitude of $k$ for simplicity; it is simple to handle the last few steps by either adjusting $k$ for the remaining steps or skipping the extrapolation.

### 4.4   ANALYSIS ON GLOBAL TRUNCATION ERRORS

We perform an analysis on global truncation errors of the Euler method and RX-Euler under the same NFEs. Assume that we solve an ODE satisfying Lipschitz condition from $t = 1$ to $t = 0$ with NFEs $= N$.

**Euler**   Since the Euler method requires a single network evaluation for each time step, the number of time steps allowed is $N$. Since the local truncation error of the Euler method on step size of $h = 1/N$ is expressed as $ch^2 + O(h^3)$, the global truncation error is given by

$$(ch^2 + O(h^3)) \cdot N = \frac{c}{N} + O(N^{-2}) = O(N^{-1}). \tag{28}$$

---

**Algorithm 1** Sampling of RX-DPM

---

**Require:** $\epsilon_\theta(\cdot)$, $N$, $T = t_N > \ldots > t_0 = 0$
1: **Input:** $k$, $\Phi(\cdot)$ (ODE solver), $p$
2: $\boldsymbol{x}_T \sim \boldsymbol{p}_T(\boldsymbol{x})$
3: **for** $i = 1$ **to** $N$ **do**
4:    *# Initialization*
5:    **if** $i \bmod k = 1$ **then**
6:       $h \leftarrow t_{N-i+1} - t_{N-i-k+1}$
7:       $\hat{\boldsymbol{x}}^{(k)}_{t_{N-i+1}} \leftarrow \boldsymbol{x}_{t_{N-i+1}}$
8:    **end if**
9:    $\lambda_i \leftarrow (t_{N-i+1} - t_{N-i})/h$
10:    $\hat{\boldsymbol{x}}^{(k)}_{t_{N-i}} \leftarrow \Phi(\hat{\boldsymbol{x}}^{(k)}_{t_{N-i+1}}, t_{N-i+1}, t_{N-i}; \epsilon_\theta(\cdot))$       *# Store $\epsilon_\theta(t)$'s if neccessary.*
11:    *# Extrapolation (Equation (22))*
12:    **if** $i \bmod k = 0$ **then**
13:       $\hat{\boldsymbol{x}}^{(1)}_{t_{N-i}} \leftarrow \Phi(\boldsymbol{x}_{t_{N-i}+h}, t_{N-i} + h, t_{N-i}; \epsilon)$       *# No NFE required.*
14:       $\tilde{\boldsymbol{x}}^{(k)}_{t_{N-i}} \leftarrow \dfrac{\hat{\boldsymbol{x}}^{(k)}_{t_{N-i}} - \sum_{j=i}^{i+k-1} \lambda_j^p \hat{\boldsymbol{x}}^{(1)}_{t_{N-i}}}{1 - \sum_{j=i}^{i+k-1} \lambda_j^p}$
15:       $\boldsymbol{x}_{t_{N-i}} \leftarrow \tilde{\boldsymbol{x}}^{(k)}_{t_{N-i}}$
16:    **end if**
17: **end for**
18: **return** $\boldsymbol{x}_{t_0}$

---

**RX-Euler**   If RX-Euler performs extrapolation every $k$ steps, the extrapolation happens $N/k$ times, where $N$ is equal to the NFEs for the Euler method. The local truncation error for RX-Euler over each $k$ steps, which has the interval of $h = k/N$, is expressed as $c'h^3 + O(h^4)$ and therefore the global truncation error is given by

$$(c'h^3 + O(h^4)) \cdot \frac{N}{k} = \frac{k^2 c'}{N^2} + O(N^{-3}) = O(N^{-2}). \tag{29}$$

RX-Euler exhibits a higher convergence rate of the global truncation error compared to the Euler method by one order of magnitude. Using the same approach, we can also demonstrate that the proposed method achieves faster convergence of the global truncation error for higher-order solvers.

## 5 EXPERIMENT

### 5.1 IMPLEMENTATION DETAILS

We conduct the experiment with EDM (Karras et al., 2022), Stable Diffusion V2[1] (Rombach et al., 2022), DPM-Solver (Lu et al., 2022), and PNDM (Liu et al., 2022) using their official implementations and provided pretrained models. Throughout all experiments, we retain the default settings from the official codebases, except for additional hyperparameters related to the proposed method. For experiments with EDM, DPM-Solver, and PNDM as backbones, we generate 50K images and compute FID (Heusel et al., 2017) using the evaluation code provided in their implementations. To evaluate Stable Diffusion V2 results, we use the PyTorch implementation for the computation of FID[2] and CLIP score[3] with the patch size of $32 \times 32$.

### 5.2 VALIDITY TEST

We first evaluate the effectiveness of RX-Euler under the EDM backbone for $k \in \{2, 3, 4\}$, where smaller $k$ values correspond to more frequent extrapolation over the same number of time steps. As

---

[1] https://github.com/Stability-AI/stablediffusion, v2-1_512-ema-pruned.ckpt
[2] https://github.com/mseitzer/pytorch-fid
[3] https://huggingface.co/openai/clip-vit-base-patch32

---

shown in Figure 2, RX-Euler consistently achieves significantly better FID scores than the Euler method across all values of $k$. In particular, when extrapolation is applied every two steps, *i.e.*, $k = 2$, our approach achieves the best performance across a wide range of NFEs. This indicates that more frequent extrapolation leads to more accurate intermediate predictions, effectively mitigating error accumulation in the final samples. Meanwhile, the curves for $k \geq 3$ remain closer to that of $k = 2$ than to the Euler method, empirically validating the reduced truncation error derived in Equation (18) for general $k$. In other words, even sparse extrapolation still has a significant impact on the output quality. For the rest of our results, we set $k = 2$.

We also compare the proposed method with conventional Richardson extrapolation, as formulated in Equation (8) with $k = 2$, which employs fixed coefficients. This baseline is labeled as Naïve ($k = 2$) in Figure 2. When comparing RX-Euler ($k = 2$) with Naïve ($k = 2$), we find that our method produces superior results. This implies that extrapolation coefficients adapted to arbitrary time step scheduling are more effective than fixed coefficients derived from uniformly discretized time steps. In other words, the proposed grid-aware coefficients enable more effective extrapolation than deterministic ones, better capturing the varying importance of precision in DPMs over time (Karras et al., 2022).

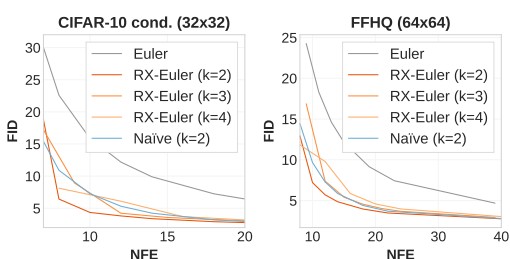

Figure 2: Effect of extrapolation on the Euler method with different $k$'s.

## 5.3 Quantitative comparisons on EDM backbone

We compare RX-Euler with other methods on four different datasets—CIFAR-10 (Krizhevsky & Hinton, 2009), FFHQ (Karras et al., 2019), AFHQv2 (Choi et al., 2020), and ImageNet (Deng et al., 2009)—using the EDM (Karras et al., 2022) backbone. Following standard practice, we evaluate the performance of class-conditional generation on CIFAR-10 and ImageNet while testing unconditional generation on the other two datasets. In this experiment, we include Heun's method, LA-DPM (Zhang et al., 2023), and IIA (Zhang et al., 2024) for comparisons with our approach, RX-Euler. Note that both Heun's method and RX-Euler are second-order numerical solvers, offering higher accuracy than the Euler method, while LA-DPM and IIA are techniques refining baseline sampling. To reproduce the results of LA-DPM, we use the Euler method, as it achieves better performance than Heun's method. For IIA (Zhang et al., 2024), we present results from the better-performing variant, selected between IIA and BIIA, as indicated in the original paper.

Figure 3 presents a comparison of FID scores across the evaluated methods over a broad range of NFEs. RX-Euler consistently outperforms the other approaches, particularly at lower NFEs, demonstrating its effectiveness in fast sampling scenarios. While it occasionally falls behind LA-DPM or IIA on CIFAR-10 within specific intervals, its overall performance remains more stable and superior across the other three datasets, which pose greater challenges due to higher resolutions and increased data diversity.

In the comparison between RX-Euler and Heun's method, while RX-Euler excels at lower NFEs, Heun's method performs slightly better at larger NFEs. This implies that selecting a more appropriate solver for each interval—between RX-Euler (extrapolation) and Heun's method (interpolation)— could yield better results. In this regard, one might expect that in the early steps—where predictions are closer to noise and thus less accurate—interpolation tends to be more stable than extrapolation. Based on this reasoning, we experiment with a hybrid approach of RX-Euler and Heun's method, labeled as RX+EDM in Figure 3. We find strong performance of this approach when RX-Euler is selectively applied to the middle or last (low-noise) few steps, outperforming both Heun's method and RX-Euler. This indicates that there is still room for improvement in our algorithm and provides another direction for future work.

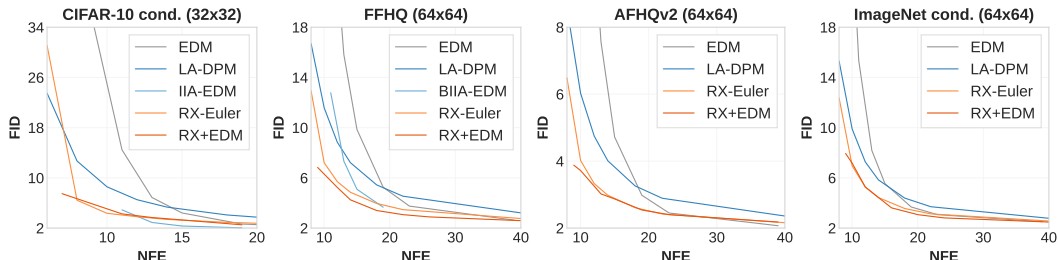

Figure 3: FIDs of RX-Euler, Heun's method (labeled as EDM), LA-DPM and IIA (or BIIA) by varying NFEs on the CIFAR-10, FFHQ, AFHQv2, and ImageNet datasets using the EDM backbone.

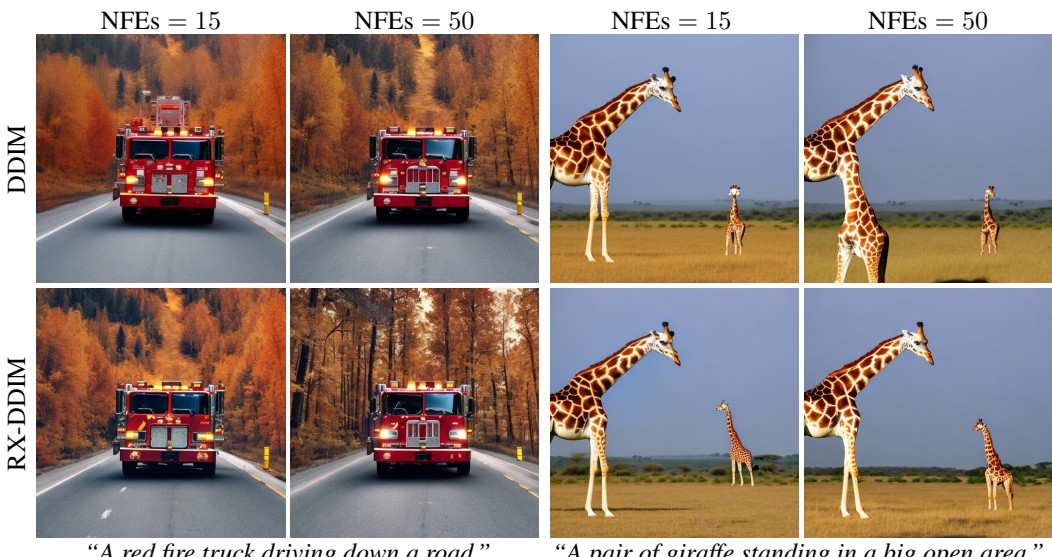

*"A red fire truck driving down a road."*  *"A pair of giraffe standing in a big open area."*

Figure 4: Qualitative results of DDIM and RX-DDIM based on Stable Diffusion V2. RX-DDIM produces sharper and more detailed backgrounds, especially evident in the left example. On the right, RX-DDIM generates more realistic giraffes, whereas DDIM struggles at NFEs = 50, failing to properly render the giraffe.

Table 1: FID and CLIP scores of DDIM and RX-DDIM using Stable Diffusion V2.

| NFEs | 15 | | 20 | | 30 | | 50 | |
|---|---|---|---|---|---|---|---|---|
| Method \ Metric | FID ($\downarrow$) | CLIP ($\uparrow$) | FID ($\downarrow$) | CLIP ($\uparrow$) | FID ($\downarrow$) | CLIP ($\uparrow$) | FID ($\downarrow$) | CLIP ($\uparrow$) |
| DDIM | 19.15 | **31.727** | 18.43 | 31.716 | 19.00 | 31.750 | 18.65 | 31.711 |
| RX-DDIM | **17.24** | 31.629 | **17.12** | **31.721** | **17.62** | **31.781** | **17.83** | **31.727** |

## 5.4 Comparisons on Stable Diffusion

We apply RX-DDIM to Stable Diffusion V2, which provides various conditional generations. For evaluation, we generate 10K $512 \times 512$ images from unique text prompts in the COCO2014 (Lin et al., 2014) validation set and compute FID and CLIP scores on resized $256 \times 256$ images. As shown in Table 1, our method also performs well on large models. However, we observe that RX-DDIM yields lower CLIP scores at NFEs = 15, which we attribute to classifier-guidance scales. According to Rombach et al. (2022), optimal classifier-free guidance scales differ across models. Since the default setting is tuned for DDIM, RX-DDIM may benefit from further optimization. Figure 4 presents qualitative comparisons between DDIM and RX-DDIM, highlighting the superior image quality of RX-DDIM. Notably, RX-DDIM generates images with more vivid colors, sharper textures, and more realistic object depictions, leading to an overall more natural appearance. We provide more examples for qualitative comparisons between DDIM and RX-DDIM in Figures 13 and 14 of Appendix F.

Table 2: FID scores of DPM-Solvers (Lu et al., 2022) and RX-DPMs applied to DPM-Solvers on CIFAR-10 and LSUN Bedroom datasets. All baseline results are reproduced under the same setting as RX-DPMs.

| Method \ NFEs | CIFAR-10 (32×32) | | | | LSUN Bedroom (256×256) | | | |
|---|---|---|---|---|---|---|---|---|
| | 9 | 10 | 12 | 15 | 9 | 10 | 12 | 15 |
| DPM-Solver-2 | – | 15.06 | 11.33 | 7.36 | – | 14.67 | 11.38 | 6.44 |
| RX-DPM-Solver-2 | – | **12.94** | **9.80** | **6.53** | – | **12.66** | **10.13** | **5.72** |
| DPM-Solver-3 | 12.39 | – | 6.76 | 5.00 | 8.79 | – | 5.37 | **4.04** |
| RX-DPM-Solver-3 | **11.50** | – | **6.62** | **4.85** | **8.12** | – | **5.18** | **4.04** |

Table 3: FID scores of two types of PNDM (Liu et al., 2022) and RX-DPMs applied to each solver on CIFAR-10, CelebA and LSUN Church datasets. Note that S-PNDM and F-PNDM require 1 and 9 additional NFEs to the number of time steps, respectively. The baseline results are copied from PNDM (Liu et al., 2022).

| Method \ # of steps | CIFAR-10 (32×32) | | | CelebA (64×64) | | | LSUN Church (256×256) | | |
|---|---|---|---|---|---|---|---|---|---|
| | 5 | 10 | 20 | 5 | 10 | 20 | 5 | 10 | 20 |
| S-PNDM | **18.3** | 8.64 | 5.77 | 15.2 | 12.2 | 9.45 | **20.5** | 11.8 | 9.20 |
| RX-S-PNDM | 19.69 | **7.64** | **4.72** | **11.56** | **9.22** | **6.89** | 21.15 | **10.96** | **8.96** |
| F-PNDM | 18.2 | 7.05 | 4.61 | 11.3 | 7.71 | 5.51 | 14.8 | **8.69** | **9.13** |
| RX-F-PNDM | – | **6.60** | **3.99** | – | **7.10** | **4.99** | – | 8.85 | 9.41 |

## 5.5 COMPARISONS ON HIGHER-ORDER SOLVERS

We further apply the proposed method to advanced ODE samplers with higher-order accuracy. Table 2 presents the effectiveness of RX-DPM when applied to DPM-Solvers (Lu et al., 2022) on CIFAR-10 and LSUN Bedroom (Yu et al., 2015). Among the variations of DPM solvers, we utilize the single-step versions of DPM-Solver-2 and DPM-Solver-3. Note that, since a single-step DPM-solver-$n$ can be considered as an $n^{\text{th}}$-order Runge-Kutta-like solver, we apply RX-DPM with $p = n + 1$ in Equation (22) for the DPM-solver-$n$. Additionally, we compare our method with another accelerated diffusion sampler, DEIS (Zhang & Chen, 2023), for class-conditioned image generation on ImageNet $(64 \times 64)$ in Table 4 of Appendix B.1. As shown in the results, RX-DPM consistently achieves the best performance across all NFEs.

As another type of advanced sampler, we consider PNDM (Liu et al., 2022). S-PNDM and F-PNDM employ linear multistep methods, specifically the 2-step and 4-step Adams-Bashforth methods, respectively, except for the initial few time steps. Accordingly, we apply RX-DPM with $p = 3$ and $p = 5$ in Equation (22) for S-PNDM and F-PNDM, respectively. The results on the CIFAR-10, CelebA (Liu et al., 2015), and LSUN Church (Yu et al., 2015) datasets are presented in Table 3. While RX-DPM improves performance in most cases, an exception arises with F-PNDM on the LSUN Church dataset, where RX-DPM does not provide a clear advantage. Upon analysis, we observe that the baseline performance of F-PNDM is highest when using 10 time steps and degrades as the number of steps increases (Liu et al., 2022). Since RX-DPM enhances accuracy by leveraging improvements in the baseline solver with finer time steps, its effectiveness is limited when the baseline itself deteriorates under finer discretization. A similar phenomenon with F-PNDM on LSUN datasets has also been reported in IIA (Zhang et al., 2024).

## 6 CONCLUSION

We introduced RX-DPM, an advanced ODE sampling method for DPMs that leverages extrapolation based on two ODE solutions derived from different discretizations of the same time interval. Our algorithm computes the optimal coefficients for arbitrary time step scheduling without additional training and incurs no extra NFEs by utilizing past predictions. This approach effectively reduces truncation errors, resulting in improved sample quality. Extensive experiments on well-established baseline models and datasets confirm that RX-DPM surpasses existing sampling methods, offering a more efficient and accurate solution for DPMs.

ACKNOWLEDGEMENTS

This work was partly supported by Samsung Research, Samsung Electronics Co., Ltd., and the Institute of Information & communications Technology Planning & Evaluation (IITP) grants [RS-2022-II220959 (No.2022-0-00959), (Part 2) Few-Shot Learning of Causal Inference in Vision and Language for Decision Making; No.RS-2021-II212068, AI Innovation Hub (AI Institute, Seoul National University); No.RS-2021-II211343, Artificial Intelligence Graduate School Program (Seoul National University)] funded by the Korea government (MSIT).

ETHICS STATEMENT

This paper utilizes pretrained diffusion probabilistic models to generate high-quality images while prioritizing efficiency. As a result, our approach does not inherently introduce hazardous elements. Nonetheless, we acknowledge the potential for unintended misuse, including the synthesis of inappropriate or sensitive content.

REPRODUCIBILITY STATEMENT

We provide detailed implementation instructions and reproducibility guidelines in Section 5.1, along with the pseudo-code of the core algorithm in Algorithm 1. The full implementation is available at https://github.com/jin01020/rx-dpm.

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

## A    JUSTIFICATION ON EQUATION (21)

Assuming a uniform grid as in the context of conventional Richardson extrapolation in Section 3.2, and the ODE solver with $O(h^{p+1})$ of local truncation error formula, we have

$$V^* = V(h) + ch^p + O(h^{p+1}) \quad \text{and} \tag{30}$$

$$V^* = V(\frac{h}{k}) + c(\frac{h}{k})^p + O(h^{p+1}) \tag{31}$$

since we expect the $O(h^p)$ of the global truncation error. Then, we have the following extrapolated solution with a truncation error of $O(h^{p+1})$ by Richardson extrapolation (Equation (8)):

$$\tilde{V}(h, k) = \frac{k^p V(h/k) - V(h)}{k^p - 1}. \tag{32}$$

Now, considering the case of Equations (20) and (21) with uniform discretization, we have

$$V^* = V(h) + c'h^{p+1} + O(h^{p+2}) \quad \text{and} \tag{33}$$

$$V^* = V(\frac{h}{k}) + kc'(\frac{h}{k})^{p+1} + O(h^{p+2}), \tag{34}$$

each correspondingly. Then, the extrapolated solution $\tilde{V}_{\text{ours}}$ obtained from solving linear system of Equations (33) and (34) becomes

$$\tilde{V}_{\text{ours}} = \frac{k^p V(h/k) - V(h)}{k^p - 1}, \tag{35}$$

which turns out to be exactly the same as Equation (32). Thus, we believe our approach can be considered to employ assumptions shared by those used in common practices of Richardson extrapolation and also can reduce global errors which is backed by experimental results as well.

## B    MORE QUANTITATIVE RESULTS

### B.1    COMPARISON WITH DEIS

We compare DEIS variants (Zhang & Chen, 2023), DPM-Solvers, and RX-DPMs applied to DPM-Solvers on class-conditioned ImageNet (64×64) in Table 4. The results clearly demonstrate that our method outperforms all other approaches across all NFEs.

### B.2    DPMs WITH OPTIMAL COVARIANCES

Although our method is designed for ODE solvers, we also conduct experiments with SN-DPM and NPR-DPM (Bao et al., 2022) on CIFAR-10 (Krizhevsky & Hinton, 2009) and CelebA datasets (Liu et al., 2015) to evaluate its performance when applied to SDE solvers. We use the official codes and provided pretrained models as described in Section 5.1. SN-DPM and NPR-DPM are two different models that correct the imperfect mean prediction in the reverse process of existing models through optimal covariance learning.

To incorporate our method into stochastic sampling, we decompose it into a deterministic sampling component and a stochastic component, and apply our method to the deterministic sampling part. Specifically, within each $k$-step interval, we execute RX-DPM algorithm using the deterministic sampling component and then add the stochasticity term afterward. In this way, our method uses the stochasticity of the baseline sampling method in a limited manner compared to the baseline sampling with the same NFEs. For a better understanding of our implementation, we provide the diagrams of the proposed method in Appendix C.

In Table 5, we show the results on NPR-DPM and SN-DPM along with the vanilla DDIM and LA-DPM (Zhang et al., 2023), which is another extrapolation-based sampling method. We observe that our method outperforms the compared methods in most cases, although a performance degradation is noted with RX-SN-DDIM on the CIFAR-10 dataset. This implies that our approach of solving the ODE might offset the benefits of the model's optimization. Despite this, we observe significant performance improvements in the most extreme case, NFEs = 10.

Table 4: Comparisons of DEIS variants, DPM-Solvers and RX-DPMs applied to DPM-Solvers on class-conditioned ImageNet (64×64). All results of DEIS and DPM-Solvers are copied from DEIS (Zhang & Chen, 2023) except for the result of DPM-Solver-3 with NFEs = 9.

| | NFEs | | | | |
|---|---|---|---|---|---|
| Method | 9 | 10 | 12 | 18 | 30 |
| tAB-DEIS | – | 6.65 | 3.99 | 3.21 | 2.81 |
| $\rho$AB-DEIS | – | 9.28 | 6.46 | 3.74 | 2.87 |
| DPM-Solver-2 | – | 7.93 | 5.36 | 3.63 | 3.00 |
| $\rho$Mid-DEIS | – | 9.12 | 6.78 | 4.00 | 2.99 |
| RX-DPM-Solver-2 | – | **6.11** | 5.61 | 3.64 | 2.93 |
| DPM-Solver-3 | 7.45 | – | 5.02 | 3.18 | 2.84 |
| $\rho$Kutta-DEIS | – | – | 13.12 | 3.63 | 2.82 |
| RX-DPM-Solver-3 | **7.08** | | **3.90** | **2.36** | **2.18** |

Table 5: FID scores for CIFAR-10 and CelebA on DDIM, NPR-DDIM and SN-DDIM models. The values for each baseline and LA-DDIM results are copied from Zhang et al. (2023).

| Dataset | CIFAR-10 (32×32) | | | CelebA (64×64) | | |
|---|---|---|---|---|---|---|
| Method \ NFEs | 10 | 25 | 50 | 10 | 25 | 50 |
| DDIM | 21.31 | 10.70 | 7.74 | 20.54 | 13.45 | 9.33 |
| RX-DDIM | **14.78** | **8.42** | **6.30** | **18.31** | **10.54** | **6.88** |
| NPR-DDIM | 13.40 | 5.43 | 3.99 | 14.94 | 9.18 | 6.17 |
| LA-NPR-DDIM | 10.74 | 4.71 | 3.64 | 14.25 | 8.83 | 5.67 |
| RX-NPR-DDIM | **6.35** | **3.92** | **3.34** | **11.58** | **6.61** | **3.98** |
| SN-DDIM | 12.19 | 4.28 | 3.39 | 10.17 | 5.62 | 3.90 |
| LA-SN-DDIM | 8.48 | **3.15** | **2.93** | 8.05 | 4.56 | 2.93 |
| RX-SN-DDIM | **7.50** | 5.12 | 4.40 | **5.20** | **2.72** | **2.25** |

## C  DIAGRAMS

Figure 5 compares the diagrams of an ODE solver, the proposed method with an ODE solver, and the proposed method with an SDE solver.

## D  COMPUTATIONAL COST

We compare the computational time and GPU memory usage of the Euler method and RX-Euler using the EDM backbone in Tables 6 and 7, respectively. For measurements, we set the batch size to 128 and use 10-step sampling on an A6000 GPU. The average runtime per batch is measured for computational time. The additional operations introduced by our method, which consist of linear combinations of precomputed values, result in negligible computational overhead compared to the time required for the network forward pass. Furthermore, as the model size increases, the relative overhead diminishes (*e.g.*, only 0.11% increase for ImageNet class-conditional sampling). Similarly, GPU memory usage increases slightly with RX-Euler, primarily due to the storage of previous predictions. However, this increase is minimal and decreases as model or data size increases, showing a similar trend to that observed with computational time.

## E  LIMITATIONS

As our method is primarily designed for an ODE solver, to integrate it with an SDE solver (or a stochastic sampling method), we partially apply the stochasticity component of the SDE solver as demonstrated in Appendix B.2. Consequently, in some cases, the effectiveness is offset because the full effects of stochasticity are not captured. However, in scenarios where NFE is very limited, which are of greater interest to us, the combined effect of RX-DPM and stochastic sampling has been empirically shown to be highly beneficial. We leave the development of methods that can perform better in more general cases for future work. Additionally, in the extension of the RX-Euler algorithm to a higher-order solver in Section Section 4.3, there remains room for improvement since we impose assumptions about linear error propagation. We believe that relaxing these assumptions or deriving more accurate equations could further enhance the performance.

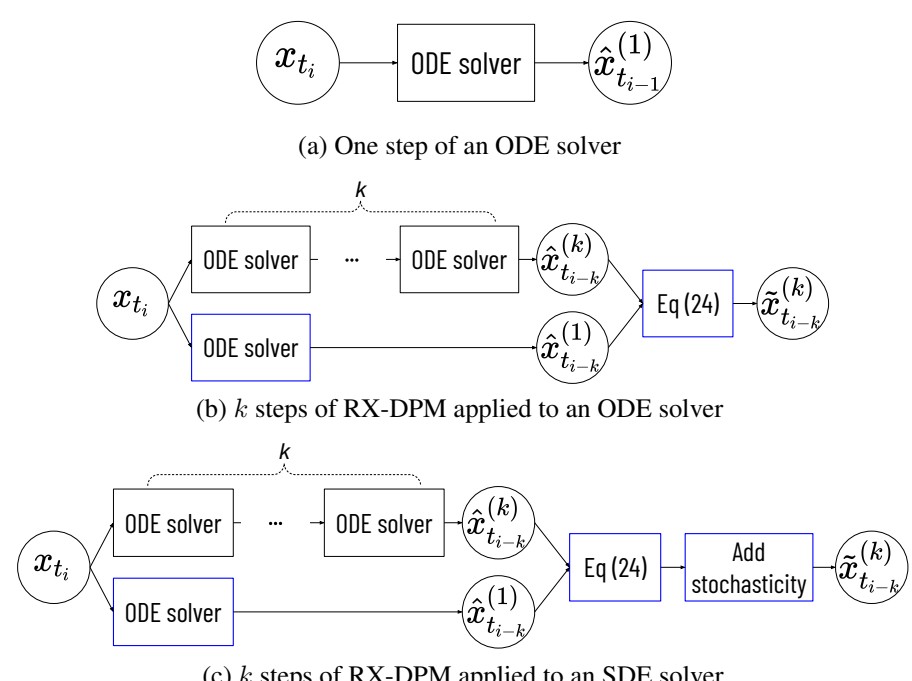

(a) One step of an ODE solver

(b) $k$ steps of RX-DPM applied to an ODE solver

(c) $k$ steps of RX-DPM applied to an SDE solver

Figure 5: Digarams of the baseline and the proposed sampling methods. The blue-bordered boxes in (b) and (c) indicate that the corresponding operation does not require network evaluation. The ODE solver in (c) refers to the deterministic sampling component of the SDE solver.

Table 6: Comparison of per-batch computation times between the Euler method and RX-Euler with the EDM backbone. The reported values represent the average runtime across 100 measurements (in seconds).

|  | CIFAR-10 cond. (32x32) | FFHQ (64x64) | ImageNet cond. (64x64) |
|---|---|---|---|
| Euler | $1.737 \pm 0.028$ | $3.895 \pm 0.023$ | $6.436 \pm 0.033$ |
| RX-Euler | $1.743 \pm 0.031$ | $3.903 \pm 0.025$ | $6.443 \pm 0.039$ |

Table 7: Comparison of GPU memory usage (MiB) during inference between the Euler method and RX-Euler with the EDM backbone.

|  | CIFAR-10 cond. (32x32) | FFHQ (64x64) | ImageNet cond. (64x64) |
|---|---|---|---|
| Euler | 2929 | 9433 | 12661 |
| RX-Euler | 3033 | 9477 | 12705 |

## F QUALITATIVE RESULTS

We provide qualitative results using the EDM backbone in Figures 6 to 12 and Stable Diffusion V2 in Figures 13 and 14.

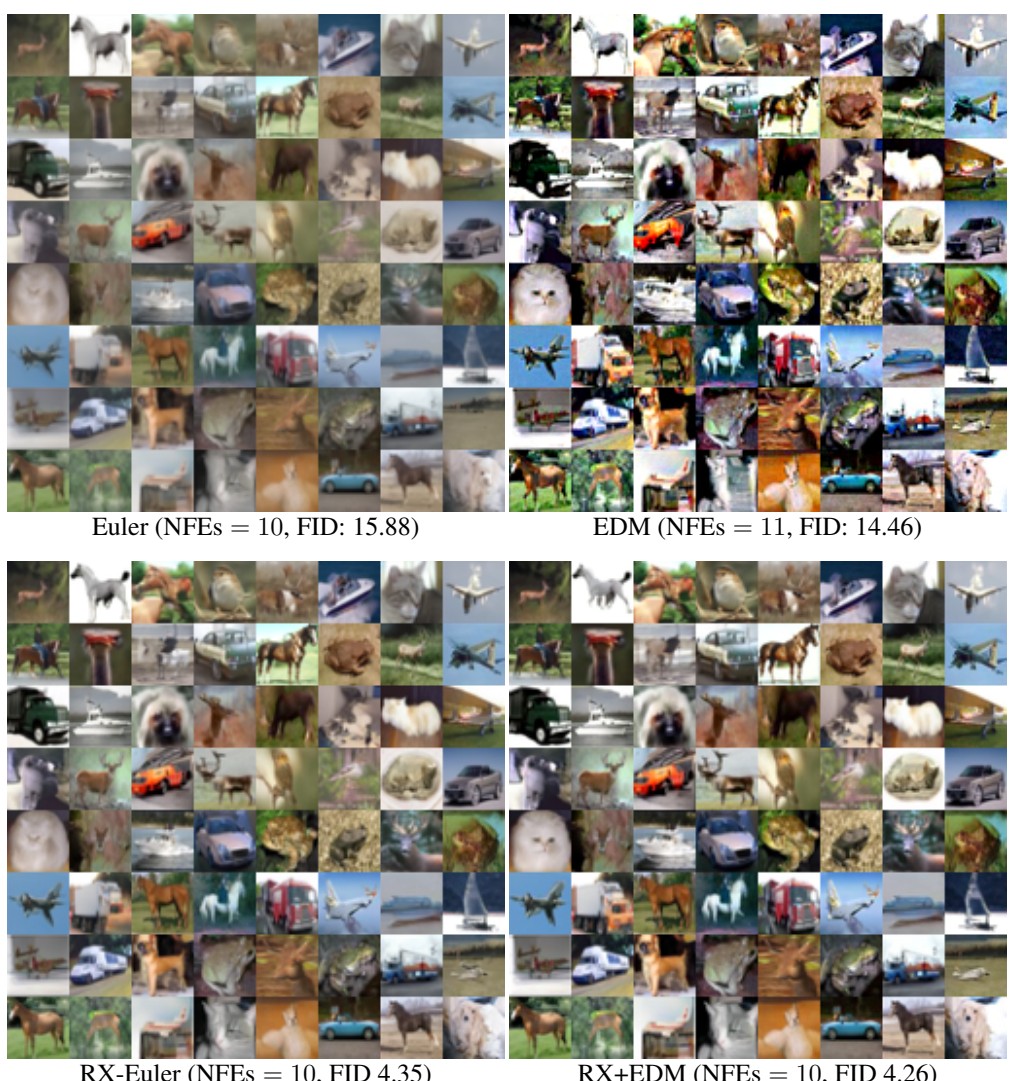

Figure 6: Qualitative results of CIFAR-10 of different sampling methods with EDM backbone.

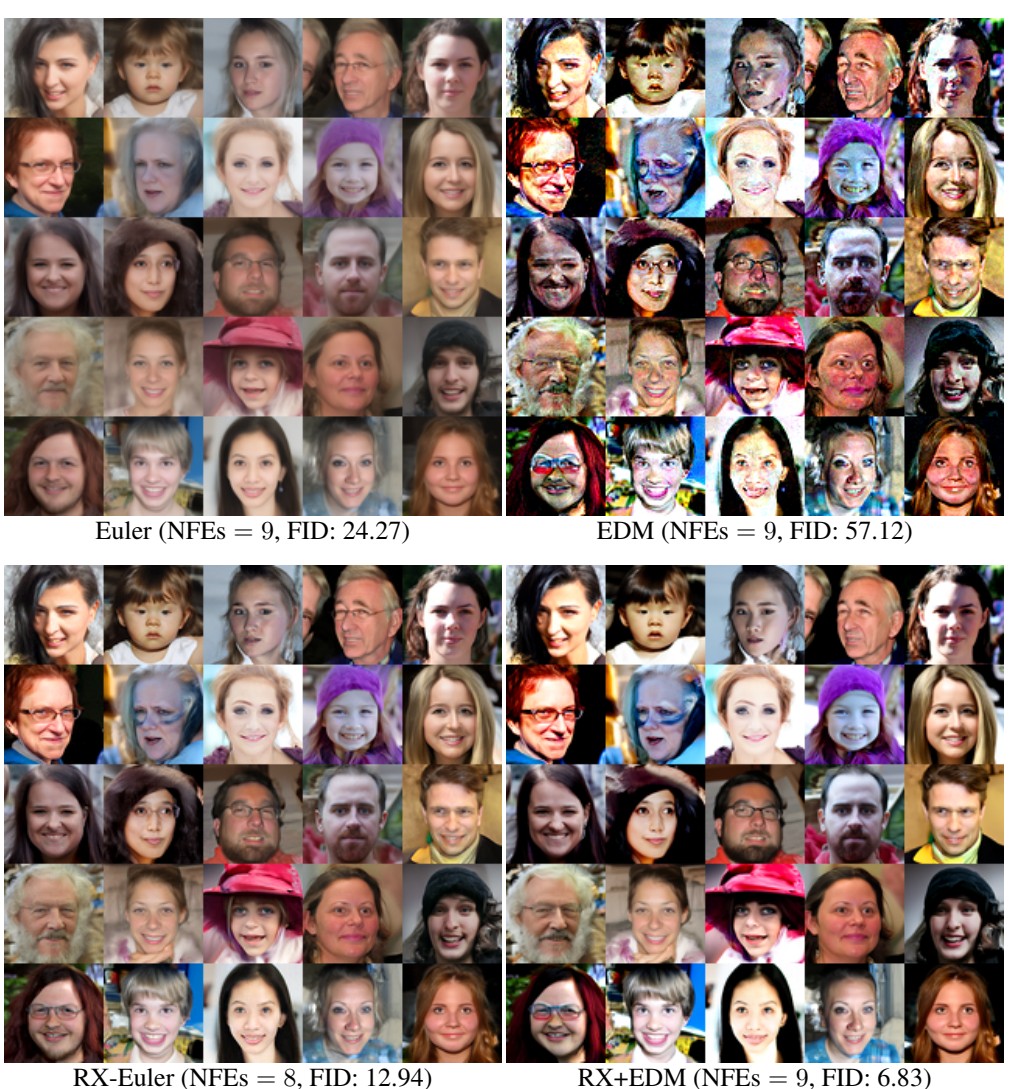

Figure 7: Qualitative results of FFHQ of different sampling methods with EDM backbone.

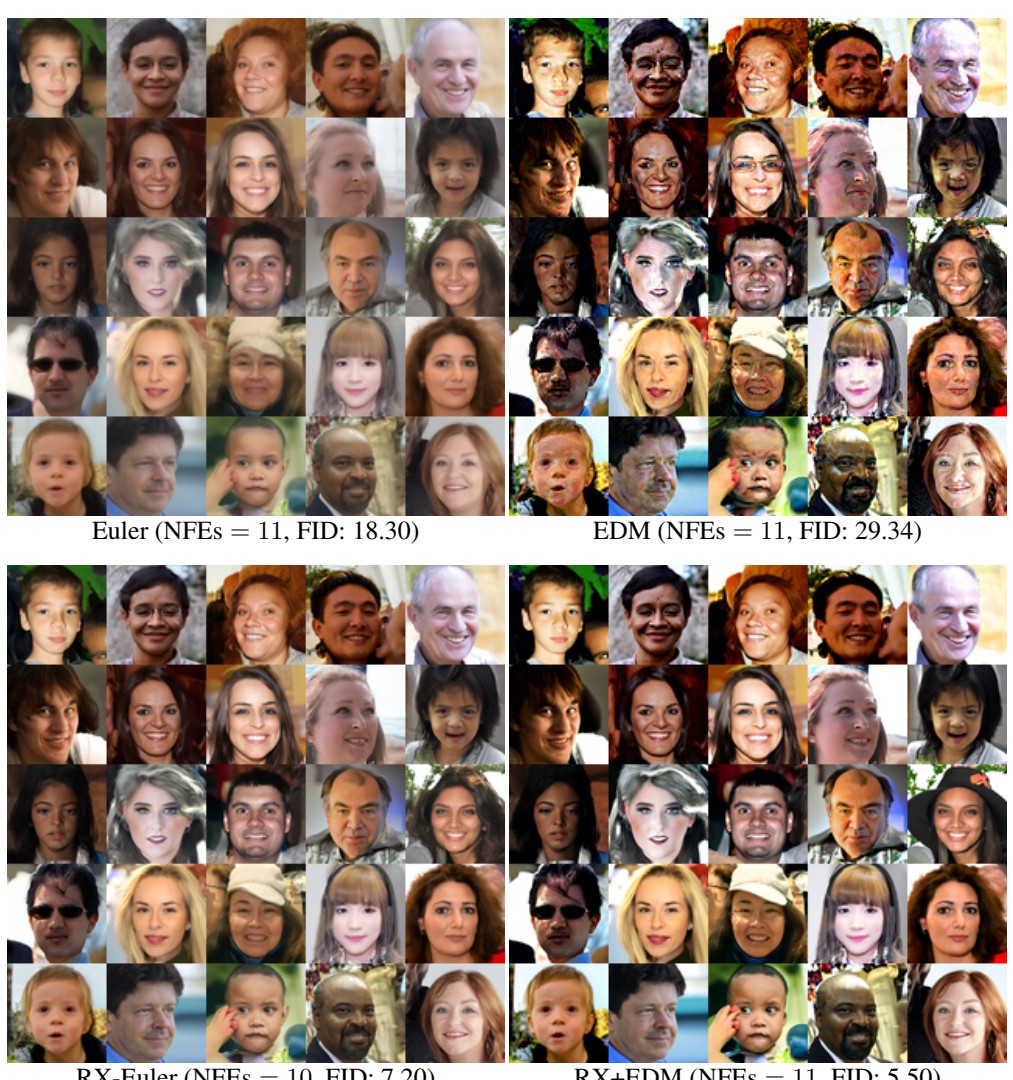

Figure 8: Qualitative results of FFHQ of different sampling methods with EDM backbone.

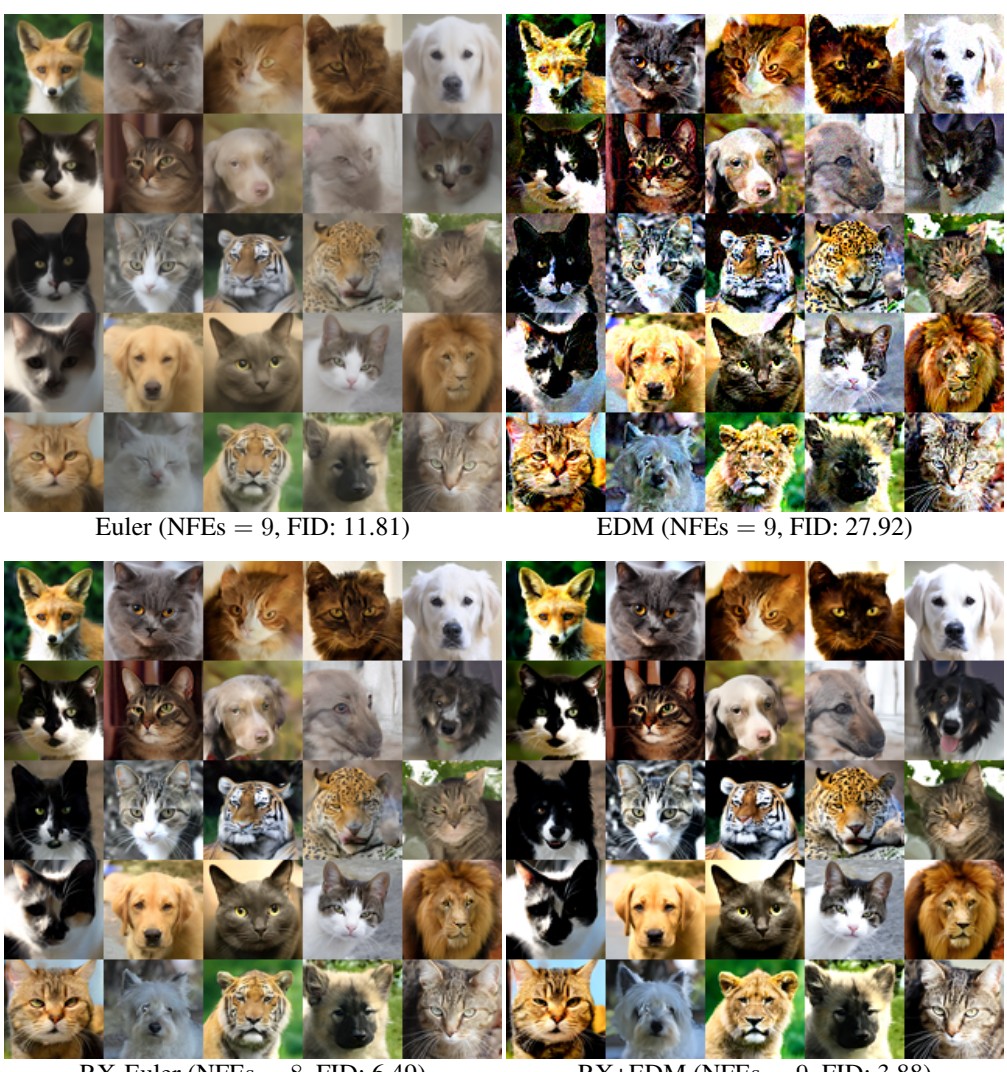

Figure 9: Qualitative results of AFHQv2 of different sampling methods with EDM backbone.

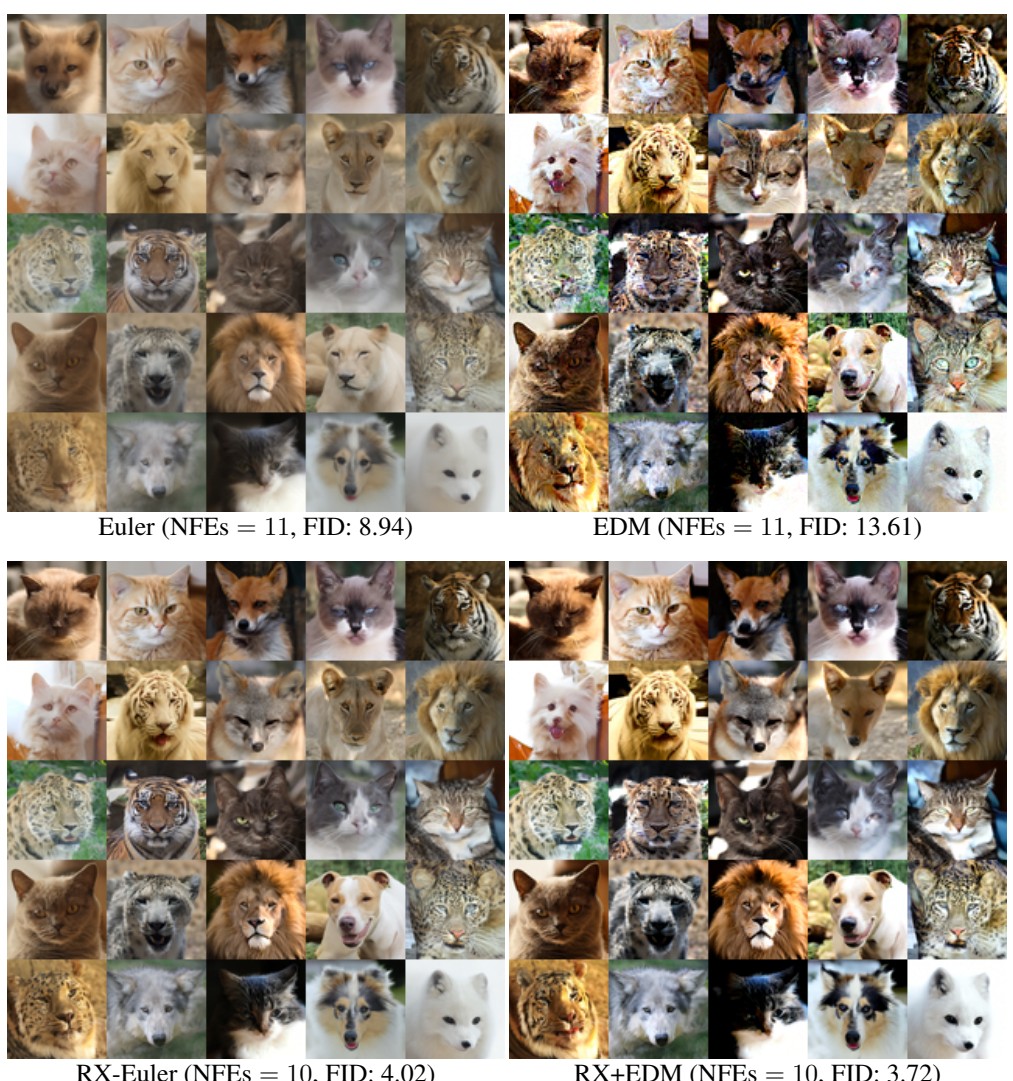

Figure 10: Qualitative results of AFHQv2 of different sampling methods with EDM backbone.

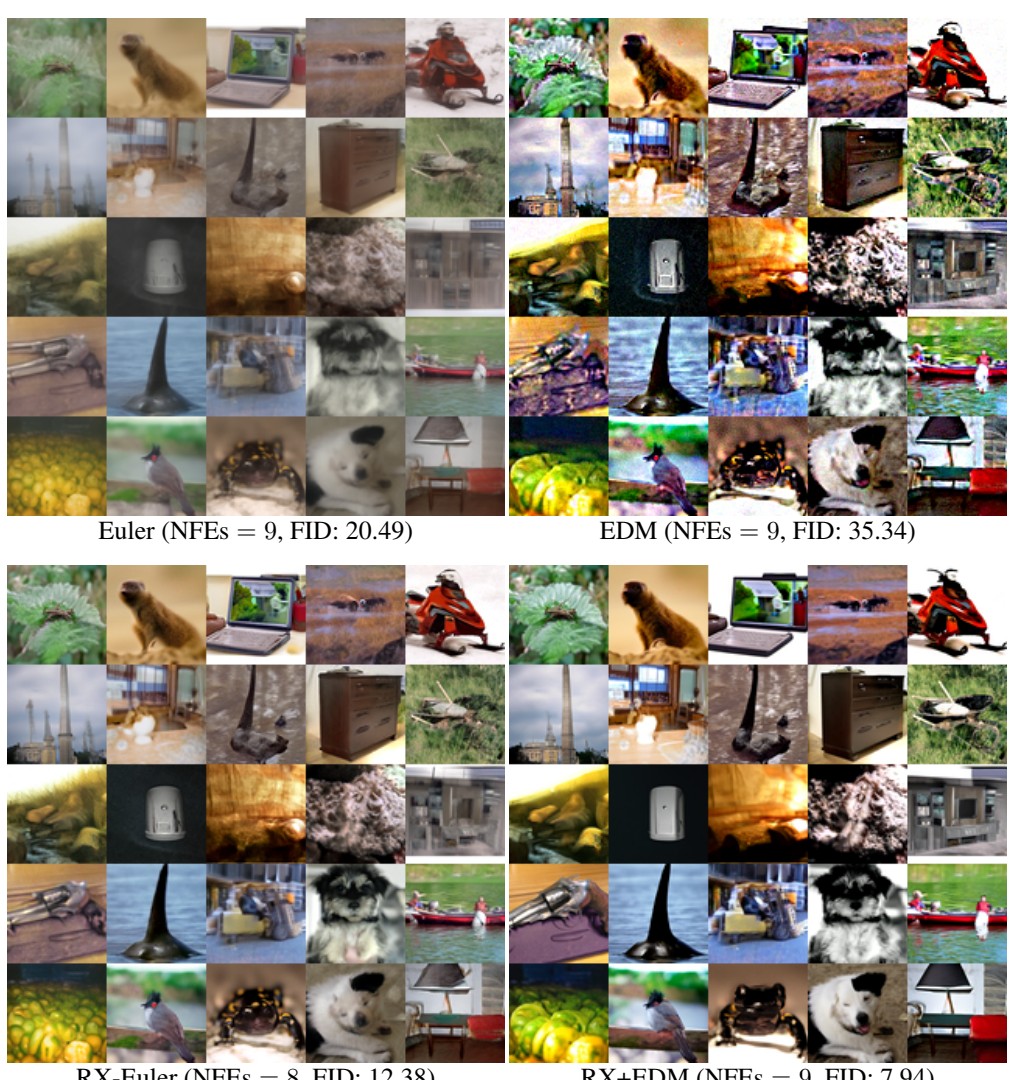

Figure 11: Qualitative results of ImageNet of different sampling methods with EDM backbone.

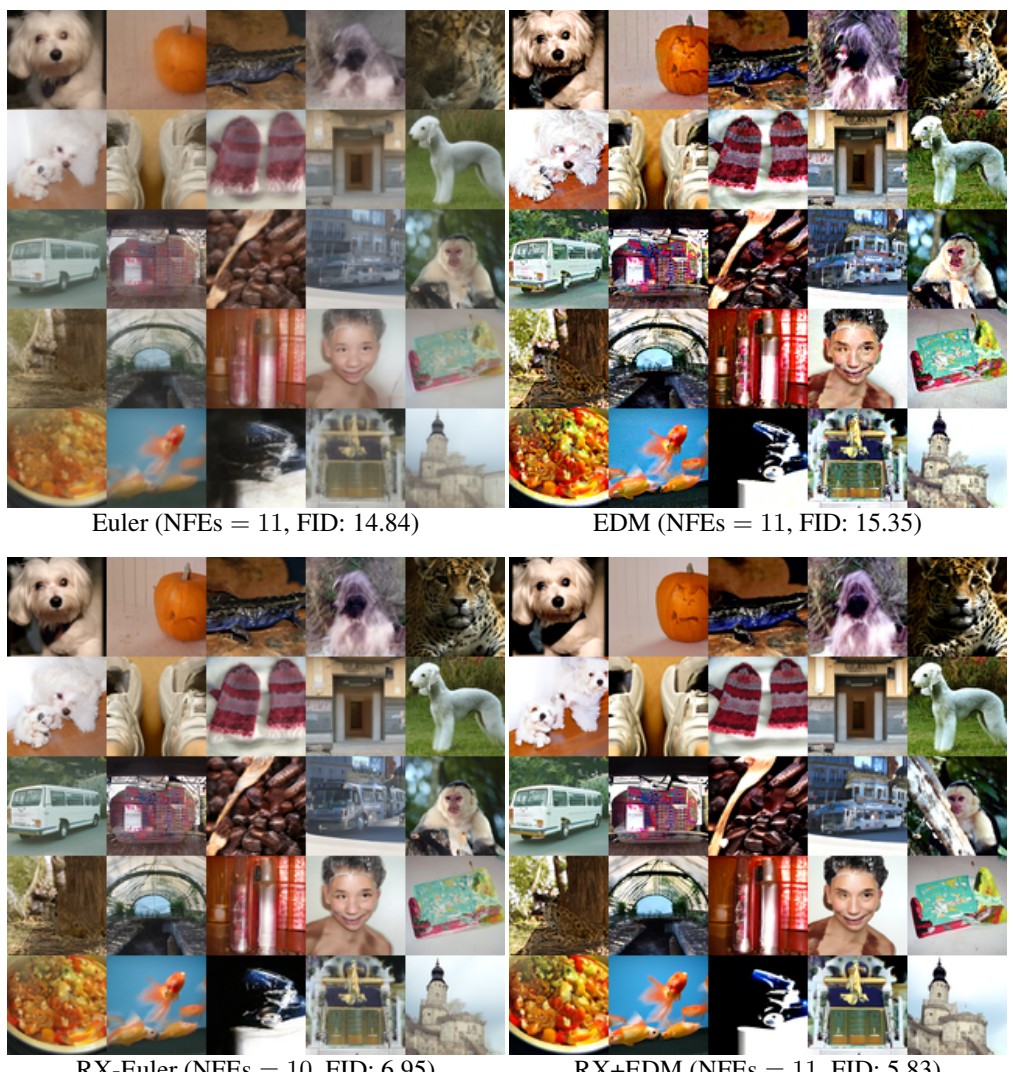

Figure 12: Qualitative results of ImageNet of different sampling methods with EDM backbone.

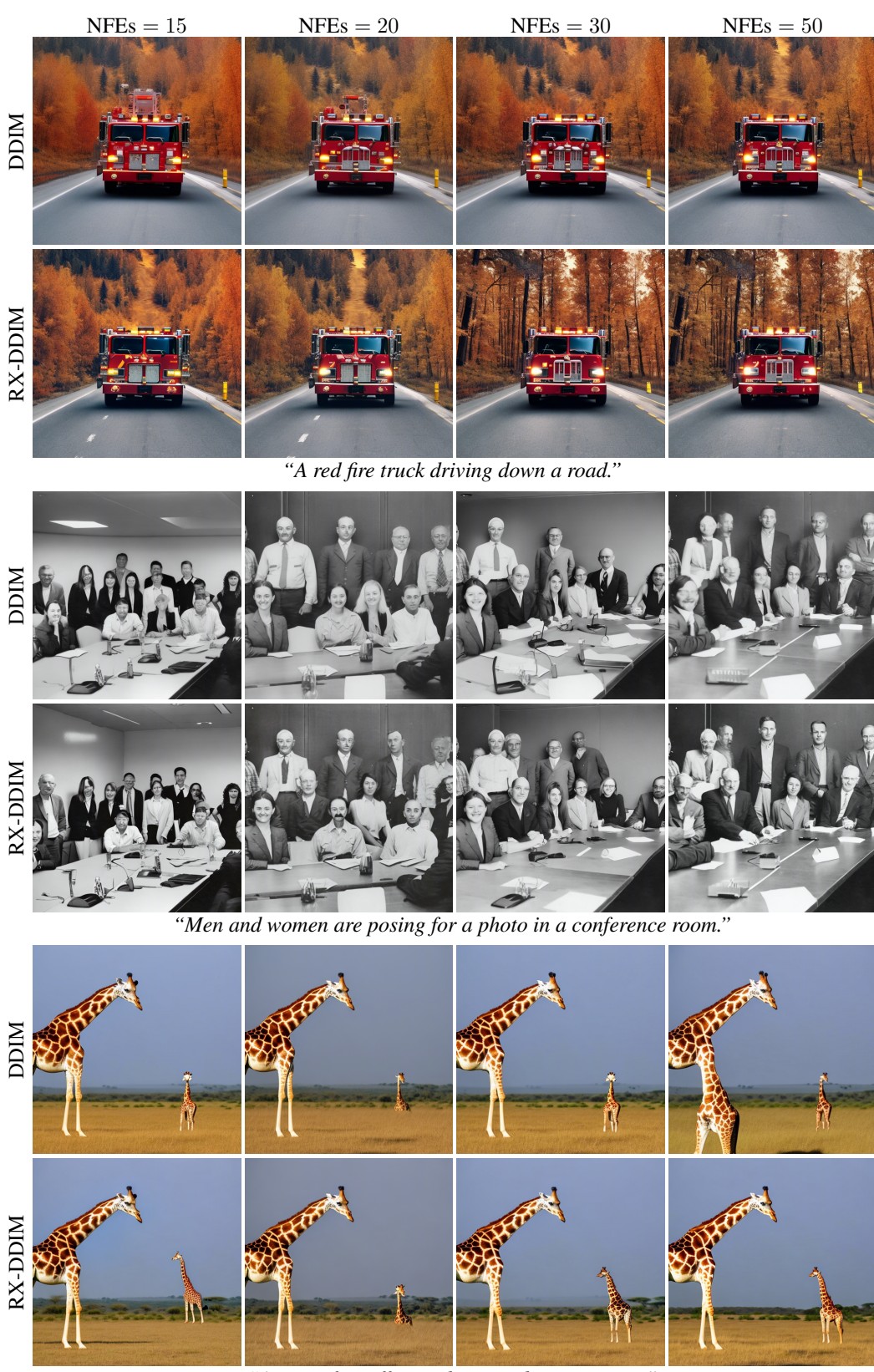

Figure 13: Qualitative results on Stable Diffusion V2 of DDIM and RX-DDIM.

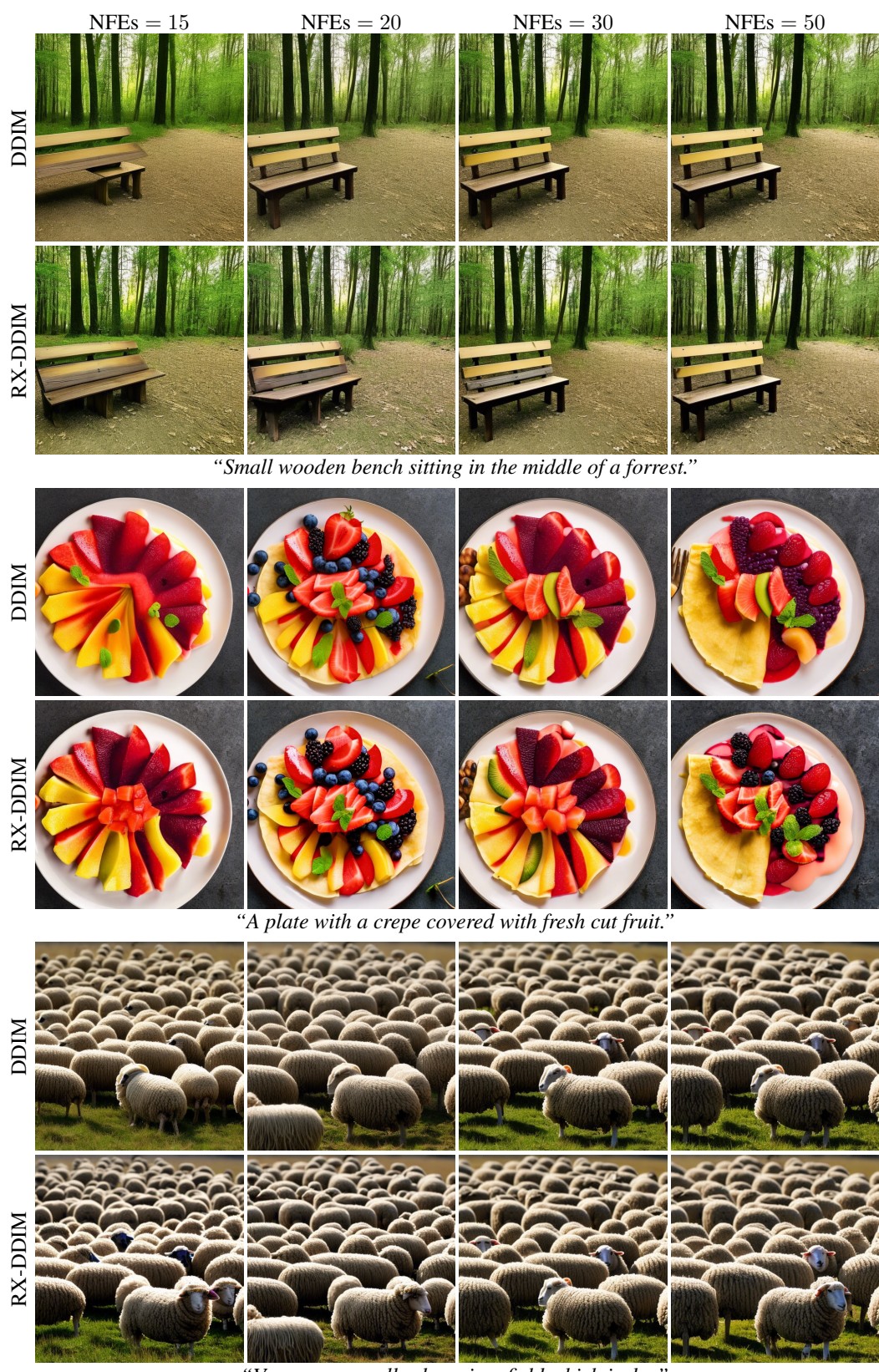

Figure 14: Qualitative results on Stable Diffusion V2 of DDIM and RX-DDIM.

