# OpenReview forum: "Enhanced Diffusion Sampling via Extrapolation with Multiple ODE Solutions"
_ICLR.cc/2025/Conference — ICLR 2025 Poster_

### Official Review · Reviewer_xBMM · 2024-10-31

**Soundness:** 2
**Presentation:** 2
**Contribution:** 2
**Rating:** 3
**Confidence:** 4

**Summary:**

The authors proposed an enhanced ODE-based sampling method for DPMs based on Richardson extrapolation. They claimed that they used non-uniform discretization approach with arbitrary time step scheduling and utilized an additional ODE solution over an interval of k time steps.

**Strengths:**

The idea of using Richardson extrapolation for ODE-based sampling method is not done in diffusion model although the idea has been explored in solving odes.

**Weaknesses:**

1. The technical part (section 4) of the paper is poorly written. I am concerned that there might be some technical errors in the formulas that lead to the main algorithm. The authors need to check carefully and they also need to refer to some existing mathematical results on using Richardson extrapolation in numerically solving odes. Also the mentioned that one of the contribution is arbitrary discretization of time steps; however it is not clear if they have actually implemented it.

2. The experiment part is not well written either. It is ambiguous on the relationship between RX-DPM  and RX-EDM and RX-DDIM,

I believe this paper needs a major revision before it is published. Please also refer to the following section - Questions.

**Questions:**

1. In page 4, line 127 p(x, \sigma) = p_data \dot N(0, sigma^2 I), the right hand side of the equation does not make sense to me.
2. I believe the local truncation error at line 196/197 in the right of eqn. (10) should not have the first term. Please check. As a result, if this is a mistake, it propagates into the following equations, and might cause major technical issue for this paper.
3. From eqn. (13) to eqn. (14) to eqn. (15), in page 4, the function f disappeared, the only reason provided was because function f is smooth. I wonder if there is an error here. They could have used the Appendix to explain the soundness of this algorithm.
4. The authors mentioned that eqn. (17) in page 5 is obtained from eqn. (16) in page 4; however, eqn. (16) has k in both subscript and superscript, one can not randomly set k =1 for superscript, and keep k in subscript; which resulting in two different values.
5. It is unclear how the authors interpret DDIM as Eqn. (20). Somehow they jumped from equation (3) to eqn. (20) without explanation.
6. In section 4.3, the authors first proposed an estimate of x_{t_{i-k}}, then jumped to RX-Runge-Kutta in page 4, when they mentioned they "consider the second order Runge-Kutta ..", but it did not mention for what reason to consider it. From the notation, I guess is they use to to estimate the truncated error. Then they jumped again into algorithm 1, in which they brought up ode solver \Phi that was not explained before. The whole section should be reorganized and be rewritten so that the algorithm can be clearly explained.
7. The paper should explicitly state that their proposed numerical solution is backward although it was time reverse in diffusion model.
8. The authors only mentioned that "We acknowledge that there are some
limitations to our method’s application in SDE ...". First of all, they proposed their algorithm for ODE, but they did not really extend it to SDE. Second, they did not state what is their limitation. Third, I would appreciate some discussion on the sensitivity of this algorithm as well as complexity cause by using the Richardson extrapolation because it is actually a more complicated way to calculate the derivative approximation.

---

> ### Author Response · Authors · 2024-11-17
> **Response to Weaknesses and Question1**
>
> Thank you for taking the time to review our work. We have addressed all the questions and concerns raised in detail and have updated the manuscript to enhance clarity and understanding. ​​If anything remains unclear, please let us know, and we will be happy to provide further clarification.
>
> ***
> > W1. The technical part (section 4) of the paper is poorly written. I am concerned that there might be some technical errors in the formulas that lead to the main algorithm. The authors need to check carefully and they also need to refer to some existing mathematical results on using Richardson extrapolation in numerically solving odes. Also the mentioned that one of the contribution is arbitrary discretization of time steps; however it is not clear if they have actually implemented it.
>
>
> We have carefully examined the formulas regarding your concerns (Q2, Q3), and confirmed that they are correct. Please refer to our responses to Q2 and Q3.
>
> We add some relevant references [1-3] of the variants of Richardson extrapolation and their applications to various problems after L50 of the original submission.
>
> Throughout Section 4, we explain the method under $k$ steps with arbitrary discretization, $[t_i, t_{i-1}, \dots, t_{i-k}]$. (Refer to L192-194 and L220-221 in the original submission).  We do not impose any assumptions on $t_i$’s. Moreover, we follow the baseline solvers’ discretizations which are usually uneven. We believe that the implementation of arbitrary discretization has been thoroughly detailed in our manuscript. However, if any concerns remain unresolved, please let us know the specific points of confusion, and we will be happy to provide clarification.
>
> [1] Shane A Richards. Completed Richardson extrapolation in space and time. Communications in numerical methods in engineering, 13(7):573–582, 1997.
>
> [2] Mike A Botchev and Jan G Verwer. Numerical integration of damped maxwell equations. SIAM Journal on Scientiﬁc Computing, 31(2):1322–1346, 2009.
>
> [3] Zahari Zlatev, Istvan Farago, and Agnes Havasi. Stability of the Richardson extrapolation applied together with the $\theta$-method. Journal of Computational and Applied Mathematics, 235(2):507–517,2010.
>
> ***
> >W2. The experiment part is not well written either. It is ambiguous on the relationship between RX-DPM and RX-EDM and RX-DDIM,
>
> RX-DPM refers to the method we propose, and any expression in the form of RX-[Solver], such as RX-DDIM, represents the application of our approach to a specific solver as an instance of RX-DPM. On the other hand, the term RX-EDM was not used in our work. In Figure 3, the label RX+EDM refers to the mixture of RX-Euler and Heun's method as discussed in L412-416 in our original manuscript. This part may have caused some confusion, and we will consider revising the naming in the future to avoid any ambiguity.
> ***
> >Q1. In page 4, line 127 p(x, \sigma) = p_data \dot N(0, sigma^2 I), the right hand side of the equation does not make sense to me.
>
> Thank you for the comment. The $\cdot$ in the equation in L137, $p(x;\sigma) = p_\text{data} \cdot \mathcal{N}(\mathbf{0}, \sigma(t)^2 \mathbf{I})$, should be changed to a convolution operator. We revised this in the revised manuscript for clarification.

---

> ### Author Response · Authors · 2024-11-17
> **Response to Questions2-5**
>
> >Q2. I believe the local truncation error at line 196/197 in the right of eqn. (10) should not have the first term. Please check. As a result, if this is a mistake, it propagates into the following equations, and might cause major technical issue for this paper.
> >Q3. From eqn. (13) to eqn. (14) to eqn. (15), in page 4, the function f disappeared, the only reason provided was because function f is smooth. I wonder if there is an error here. They could have used the Appendix to explain the soundness of this algorithm.
>
> Eq.(10) is correct. This is also discussed in Sec 211 (p60) of [4]. Intuitively, the equation means the estimation by the Euler method, $\\hat x_{t_{i-1}}^{(1)}$, for the exact solution $x_{t_{i-1}}^*$ has a quadratic local truncation error.
>
> We also have confirmed that the derivation from Eq.(13) to Eq.(15) has no problem. We provide a more detailed derivation for better understanding below:
>
> Using Eq.(10), the first two terms of Eq.(13) satisfy the following relationship:
> $$
> x_{t_{i-1}}^* - \\lambda_2hf(x_{t_{i-1}}^*) = x_{t_{i-2}}^* - \\frac{1}{2}x_{t_{i-1}}^{*''}\\lambda_2^2h^2 + O(h^3).
> $$
> In general, a function $g$ is considered smooth if $g \in C^k$ for $k \geq 3$, i.e., it is $k$-times continuously differentiable. Consequently, in this work, a smooth function $f$ is locally Lipschitz continuous. Leveraging Eq.(10) and the local Lipschitz continuity of $f$, the last two terms of Eq.(13) satisfy:
>
>
> $$
> \\begin{align}
> -\\lambda_2hf(\\hat x_{t_{i-1}}^{(1)}) + \\lambda_2hf(x_{t_{i-1}}^*) \\
> = \\lambda_2 h (f(x_{t_{i-1}}^*) - f(x_{t_{i-1}}^* + O(h^2))) \\
> = \\lambda_2h O(h^2) = O(h^3).
> \\end{align}
> $$
>
> Combining the two equations above, Eq.(14) follows directly from Eq.(13). Furthermore, $x^{''} = f^{'}$ is also locally Lipschitz continuous. This yields:
>
> $$
> x_{t_{i-1}}^{*''} = f'(x_{t_{i-1}}^{\*},  t_{i-1}) \\
> = f^{'}(x_{t_i} + O(h), t_{i} -h) ~~(\\because \\text{Eq (10)})\\
> $$
> $$
> = f^{'}(x_{t_i},t_i) + O(h) ~(\\because \\text{Lipshitz continuity of $f^{'}$}) \\
> $$
> $$
> = x_{t_i}^{''} + O(h)
> $$
> Finally, we obtain Eq.(15) by applying the above result in Eq.(14).
>
> [4] Butcher, John C. (2003). Numerical Methods for Ordinary Differential Equations. New York: John Wiley & Sons. ISBN 978-0-471-96758-3.
> ***
> >Q4. The authors mentioned that eqn. (17) in page 5 is obtained from eqn. (16) in page 4; however, eqn. (16) has k in both subscript and superscript, one can not randomly set k =1 for superscript, and keep k in subscript; which resulting in two different values.
>
> The general notation for a numerical solution, $\\hat x_{t_{j}}^{(n)}$, indicates that the solution at $t_j$ is obtained after $n$ iterations. Specifically, the subscript represents the diffusion time step of the solution, while the superscript $n$ denotes the number of iterations performed by the numerical solver. Eq.(17) and Eq.(18) are specific instances of $\\hat x_{t_{j}}^{(n)}$, where for Eq.(17), $j = i - k$ and $n = 1$, and for Eq.(18), $j = i - k$ and $n = k$. Although the notation is well-defined, the previous text may lead to ambiguity for readers. To address this, we have clarified in L230-231 of original submission that Eq.(17) is derived from Eq.(10), and Eq.(18) is derived from Eq.(16). Additionally, we provide a general definition of $\hat x_{t_j}^{(n)}$ in L194-196 of original submission for further clarity.
> ***
> >Q5. It is unclear how the authors interpret DDIM as Eqn. (20). Somehow they jumped from equation (3) to eqn. (20) without explanation.
>
> The relevance between DDIM and neural ODE is discussed in DDIM [5]. Eq.(20) simply adopts their results with slightly different notations.
>
> [5] Jiaming Song, Chenlin Meng, and Stefano Ermon. Denoising diffusion implicit models. In ICLR, 2021

---

> ### Author Response · Authors · 2024-11-17
> **Response to Questions6-7**
>
> >Q6. In section 4.3, the authors first proposed an estimate of x_{t_{i-k}}, then jumped to RX-Runge-Kutta in page 4, when they mentioned they "consider the second order Runge-Kutta ..", but it did not mention for what reason to consider it. From the notation, I guess is they use to to estimate the truncated error. Then they jumped again into algorithm 1, in which they brought up ode solver \Phi that was not explained before. The whole section should be reorganized and be rewritten so that the algorithm can be clearly explained.
>
> In the original submission, Section 4.3 consists of three parts. First, we present how we obtain an extrapolation equation for arbitrary ODE solvers with local truncation error order of $p$ (L257-273). Second, we give examples of obtaining one-step estimation, $\hat x_{t_{i-k}}^{(1)}$, for higher-order solvers (L273-302). Finally, Algorithm 1 explains the whole process of RX-DPMs for arbitrary ODE solvers (L302-305). The first and second parts correspond to the 12th and 11th row of Algorithm 1, respectively. Meanwhile, we have provided a more detailed explanation and reorganized Section 4.3 to enhance clarity and facilitate better understanding in the revised manuscript.
>
> The reason why we consider the second order Runge-Kutta is to explain RX-DPMs with specific examples as mentioned in L273-274 of the original submission. We would like to clarify that it is not to estimate truncation error as the reviewer questioned; the truncation error equations of the solvers are already well-known. We provide two paragraphs, RX-Runge-Kutta and RX-Adam-Bashforth, where each example shows how to apply the proposed method to the popular ODE solver families. Particularly, we discuss how we can obtain one-step estimation $\hat x_{t_{i-k}}^{(1)}$  for higher-order solvers without additional network evaluations. The reason why we explain with the specific example as the second-order Runge-Kutta method is that obtaining $\hat x_{t_{i-k}}^{(1)}$ without an increase in NFE is not obvious and may vary across solvers. Therefore, we explained our methods for widely used solvers for diffusion models to cover as many solvers as possible. For instance, one of the baseline solvers, DPM-Solver-2, can be considered as a second-order Runge-Kutta method.
>
> ***
> > Q7. The paper should explicitly state that their proposed numerical solution is backward although it was time reverse in diffusion model.
>
> Thank you for the suggestion. We explicitly mentioned that the solution is obtained in the time-reversed direction in L192 of the original submission.

---

> ### Author Response · Authors · 2024-11-17
> **Response to Question8**
>
> >Q8. The authors only mentioned that "We acknowledge that there are some limitations to our method’s application in SDE ...". First of all, they proposed their algorithm for ODE, but they did not really extend it to SDE. Second, they did not state what is their limitation. Third, I would appreciate some discussion on the sensitivity of this algorithm as well as complexity cause by using the Richardson extrapolation because it is actually a more complicated way to calculate the derivative approximation.
>
>
> To incorporate our method with SDE solver (or stochastic sampling method), we interpret an SDE solver as a combination of a deterministic sampling component and a stochasticity component, and we utilize the deterministic sampling part as an ODE solver. Specifically, within each $k$-step interval (the basic unit where extrapolation occurs), we execute the RX-DPM algorithm using deterministic sampling taken from the SDE solver and add stochasticity term afterwards. Consequently, we leverage the effect of accurate mean prediction of RX-DPM while partially incorporating the mean correction effect that stochasticity provides. This effect can vary depending on the characteristics of the trained models and datasets. When the mean correction from stochasticity is large and significant, there may be some offsetting effect with our method which is stated as a limitation. However, when the mean correction effect from stochasticity is relatively small, our method harmonizes effectively with SDE sampling, producing excellent results.
> As an illustrative example, Section 5.6 explores NPR-DDIM and SN-DDIM, which propose to learn optimal variances applicable to the deterministic DDIM. Please note that we have detailed the implementation and discussed the limitations through the analysis of the results of Table 4 in our original submission. We have also added diagrams of the proposed methods in Appendix C in the revised manuscript for a more intuitive understanding.
>
>
> Regarding the sensitivity of RX-DPM, we have applied our algorithm to numerous ODE solvers such as Euler method, DDIM, DPM-Solver-2/3, and PNDMs on various datasets. RX-DPM enhances sample quality for most cases, and it does not incorporate any hyperparameter except $k$, which controls the frequency of extrapolation. This implies that RX-DPM is a universal method.
>
>
> Moreover, the additional operations performed by our algorithm compared to existing methods are computing the 1-step estimation for the $k$-step interval (L11 of Algorithm 1) and performing extrapolation (L12 of Algorithm 1). Although the equations may appear complex, they consist solely of linear combinations of pre-calculated values. Therefore, RX-DPM is easy to implement and does not require additional network evaluations or significant overhead.
> We also provide a comparison of the computation complexity between the Euler method and RX-Euler using the EDM backbone in the table below, which demonstrates that our method incurs negligible computational overhead. This content has been included with further details in Appendix E of the revised manuscript.
>
> __Comparison of average runtime / batch (seconds)__
>
> |          | CIFAR-10 cond. (32x32) |   FFHQ (64x64)   | ImageNet cond. (64x64) |
> |----------|:----------------------:|:----------------:|:----------------------:|
> | Euler    |    1.737 $\\pm$ 0.028    | 3.895 $\\pm$ 0.023 |    6.436 $\\pm$ 0.033    |
> | RX-Euler |    1.743 $\\pm$ 0.031    | 3.903 $\\pm$ 0.025 |    6.443 $\\pm$ 0.039    |

---

> > ### Comment · Reviewer_xBMM · 2024-11-21
> > **Overall reply to the authors**
> >
> > Thanks for the authors prompt response! However, I feel my concerns are not fully addressed. Overall, this paper actually proposed an extrapolation method based on the existing diffusion model algorithms. However, there are more recent fast processing algorithms. For example, the consistency model algorithm may already has fast one-step generation and excellent reconstruction directly mapping noise to data. This seems to make this proposed extrapolation method unnecessary, only added processing time (although relatively small overhead) as we can see from the table the authors provided.
> >
> > Specifically, I still believe this paper has major technical error that need the authors to address carefully. Obvious it begins with something wrong with the definition of the local truncation error given in eqn. (10).  The authors should double check reference [4] they provided. In addition, it leads to eqns. (19) and (23) (in the resubmission). The extrapolation formula ignored derivatives (1st and 2nd order ) of the function, which I believe it only fits a function that meets very specific requirements.
> >
> > While the update in the new submission is not enough to address my concerns, after I went back to review the paper again, I also noticed the  Eqn. (20) is poorly explained and I found that it is difficult to associated it to the reference provided. Thus, overall, it is still not that clear to me on how this proposed algorithm works.
> >
> > Also, in the updated resubmission, the authors did not address any limitation lead to the reconstruction error that can be observed from a picture. For example, both EX-methods they applied led (or inherited from the methods they used for extropolation) to two different kinds of pumpkin (1st row) and Monkey (3rd row) and in Figure 10.
> >
> > Based on the above comments, I chose to keep my original score.

---

> > > ### Author Response · Authors · 2024-11-29
> > >
> > > Dear reviewer xBMM,
> > >
> > > We kindly request review on our responses, especially regarding the clarification on equations that you have doubted.
> > > Meanwhile, reviewer 6Qbz has expressed that there are no problems with the equations in the paper.
> > > We would appreciate your feedback on any aspects you have questioned.
> > >
> > > Best regards,
> > >
> > > authors

---

> > > > ### Comment · Reviewer_xBMM · 2024-11-30
> > > >
> > > > Thanks to the authors response!
> > > >
> > > > The responses from the authors confirmed my concern that their extrapolation algorithm inherited errors from the baseline sampling methods that they used for extrapolation. Meanwhile,  DPM-Solver-k DPM utilized the idea that x_t near x_s can be simplified into a very special exact formulation, which can be approximated by  applying Taylor expansion to achieves O(h^k);  In this submission, while the authors claimed that their truncation error of O(h^2), however, there is no detailed discussion to address this potential issue related to DPM Solvers.
> > > >
> > > > Meanwhile I am still concerned potential issues with this algorithm, and whether the local Lipschitz continuity of f and its derivative is too strong.
> > > >
> > > > In regard to the definition of the local truncated error. In Page 4, the authors stated that the local truncation error of .... is given {\hat{x}}^{(1)}_{t_{i-1}} while defining this as the numerical solution; this will confuse reader. We define error between x_1 and x_0 as error = x_1 - x_0; to me, it is unacceptable to say the error defined as x1 because this obviously confuses readers.
> > > >
> > > > The authors still did not pinpoint the reference used for Eqn. (20) and I hope they have revised their paper to make it more easier for the readers. I understand this is re-parameterization of diffusion model, however, I am concerned if there is error in the y form. The reference was not correct in the paper as well.

---

> > > > > ### Author Response · Authors · 2024-12-03
> > > > > **Summary on major points**
> > > > >
> > > > > As the review process nears its conclusion and numerous discussions have been made, we would like to summarize two major issues discussed so far and how we have addressed them.
> > > > >
> > > > > First, we would like to emphasize once again that there is no issue with the validity of Eqs (10, 13, 14, 15, 19, 23), which were initially raised to have technical flaws. These equations have been thoroughly explained and proven correct during the discussion period. It seems that the remaining concerns are no longer related to technical issues, as the reviewer’s latest response focuses only on readability.
> > > > >
> > > > > Second, regarding the doubts raised about the DDIM ODE equation, Eq. (20), we have clarified the exact location in the reference and explained the differences in notation, conclusively proving that the equation is correct.
> > > > >
> > > > > Therefore, we believe there is no issue with the validity of these equations, and the experiments have been properly implemented. We hope the reviewer now fully understands and any misunderstandings regarding the validity of these equations have been resolved.

---

> ### Author Response · Authors · 2024-11-22
> **Responses (1/2)**
>
> Thank you for the response. We provide our answers to further questions and concerns below, hoping that they clear out any remaining doubts.
>
> ---
>
> >Overall, this paper actually proposed an extrapolation method based on the existing diffusion model algorithms. However, there are more recent fast processing algorithms. For example, the consistency model algorithm may already has fast one-step generation and excellent reconstruction directly mapping noise to data. This seems to make this proposed extrapolation method unnecessary, only added processing time (although relatively small overhead) as we can see from the table the authors provided.
>
> We respectfully disagree with the opinion that one-step generation models render sampling acceleration methods unnecessary. We would like to emphasize that our approach is not directly comparable to such distillation-based one-step generation models. While distillation-based methods require additional model training, our algorithm is a training-free sampling method that can be applied to any ODE-based sampling technique.
>
> In this sense, unlike consistency models, our approach preserves the original diffusion model's flexibility, allowing for quality adjustment based on the sampling budget. This key advantage makes the method flexible to different needs. Moreover, research on generative models with iterative sampling schemes is still actively ongoing. Our sampling method can also be applied to various tasks, including image-to-image translation, which has inspired numerous studies exploring diverse approaches leveraging the iterative sampling process of diffusion models.
>
> Furthermore, we believe that our method can also serve as an enhanced teacher for distillation-based approaches. Many distillation methods rely on ODE samplers during training (e.g., consistency models achieve improved performance when trained with Heun’s method compared to Euler’s method). Therefore, combining our approach with distillation-based methods has the potential to yield superior results.
>
> Finally, sampling acceleration methods that utilize pretrained models remain an active line of research, with recent studies proposing high-order solvers or improving existing sampling methods through numerical techniques [6-8] (also discussed in L113–129). Our method falls within this category while introducing a distinct approach that sets it apart from previous methods.
>
> These factors strengthen our belief that such approaches are far from obsolete and our method should be independently considered from one-step generation models.
>
> [6] Zhang, Q., and Chen, Y., Fast sampling of diffusion models with exponential integrator, In ICLR, 2023.
>
> [7] Zhang, G. et al., Lookahead diffusion probabilistic models for refining mean estimation, In CVPR, 2023.
>
> [8] Zhang, G. et al., On accelerating diffusion-based sampling process via improved integration approximation, In ICLR, 2024.
>
> ---
>
> >I still believe this paper has major technical error that need the authors to address carefully. Obvious it begins with something wrong with the definition of the local truncation error given in eqn. (10). The authors should double check reference [4] they provided. In addition, it leads to eqns. (19) and (23) (in the resubmission). The extrapolation formula ignored derivatives (1st and 2nd order ) of the function, which I believe it only fits a function that meets very specific requirements.
>
> As stated in our initial response, there is no error in Eq. (10). It simply applies the well-known local truncation error formula, the difference between the exact solution and the single-step approximation, $x_{t_{i-1}}^* - \\hat x_{t_{i-1}}^{(1)}$, of the Euler method. The proof of this can be easily derived using Taylor's expansion, as shown in the reference we provided [4]. If the reviewer's confusion arises from not starting with the definition of LTE (local truncation error) on the left-hand side, rewriting the left-hand side as $x_{t_{i-1}}^* - \\hat x_{t_{i-1}}^{(1)}$ from Eq.(10) still remains self-evidently correct as follows, contrary to the reviewer's belief.
> $$
> x_{t_{i-1}}^{*} - \\hat x_{t_{i-1}}^{(1)} = \\frac{1}{2} x_{t_i}'' \\lambda_1^2 h^2 + O(h^3).
> $$
>
>
> Eq. (19) and Eq. (23) are derived by solving a linear system of two equations. Specifically, Eq. (19) is obtained by eliminating the term involving $h^2$ of Eq.(17) and (18), while Eq. (23) results from eliminating the term involving $h^p$ of Eq. (21) and (22). The derivative terms in (10) and (16) are treated as constants in (17) and (18) because $x_{t_i}$ is given, making its second derivative constant. Since these terms are ultimately eliminated in (19), their exact values are unimportant. By leveraging the two equations of the ODE solutions, we eliminate these derivative-related terms to construct a higher-accuracy approximation. This approach is one of the key techniques in Richardson extrapolation, which is also explained in Sec 3.2.

---

> ### Author Response · Authors · 2024-11-22
> **Responses (2/2)**
>
> >While the update in the new submission is not enough to address my concerns, after I went back to review the paper again, I also noticed the Eqn. (20) is poorly explained and I found that it is difficult to associated it to the reference provided. Thus, overall, it is still not that clear to me on how this proposed algorithm works.
>
> Eq. (14) of DDIM [5] and Eq. (20) of our paper are clearly identical, differing only in notation: $\gamma$ instead of $\sigma$ and $y$ instead of $\bar x$. DDIM [5] also states that "can be treated as a Euler method over the following ODE" above Eq. (14) in their paper.
>
> [5] Jiaming Song, Chenlin Meng, and Stefano Ermon. Denoising diffusion implicit models. In ICLR, 2021
>
> ---
>
> >The authors did not address any limitation lead to the reconstruction error that can be observed from a picture. For example, both EX-methods they applied led (or inherited from the methods they used for extropolation) to two different kinds of pumpkin (1st row) and Monkey (3rd row) and in Figure 10.
>
> The differences observed between RX-Euler and RX+EDM are not due to extrapolation but rather stem from the baseline sampling methods. In Figure 10, the bottom-left side shows RX-Euler, which applies our algorithm to the Euler method, while the bottom-right side combines RX-Euler with EDM (Heun’s method). Consequently, the image details may vary depending on the baseline sampling methods, which is fundamentally a natural phenomenon. Therefore, in Figures 7-10, the focus should be on observing that the bottom two methods outperform the top two methods. As such, comparing the two bottom methods, RX-Euler and RX+EDM, should not be considered a limitation.

---

> ### Author Response · Authors · 2024-11-30
> **Response (1/2)**
>
> Thank you for your response. We have provided detailed explanations for each of your points below.
>
> >​​The responses from the authors confirmed my concern that their extrapolation algorithm inherited errors from the baseline sampling methods that they used for extrapolation.
>
> The error itself is not inherently problematic, as numerical solutions to ODEs/SDEs are always subject to some degree of numerical error. The key focus, however, should be on the magnitude of these errors during sampling. Our method aims to mitigate errors from the baseline sampling through extrapolation, thereby improving the quality of the generated samples. We would appreciate it if the reviewer could clarify which specific part of our response suggests that the errors are being inherited, as we are currently having difficulty identifying the basis for the concerns regarding errors in the equations.
>
> ---
> >Meanwhile, DPM-Solver-k DPM utilized the idea that x_t near x_s can be simplified into a very special exact formulation, which can be approximated by applying Taylor expansion to achieves O(h^k); In this submission, while the authors claimed that their truncation error of O(h^2), however, there is no detailed discussion to address this potential issue related to DPM Solvers.
>
> Firstly, DPM-Solver-n can be thought of as a type of $n$-th order Runge-Kutta solver. This implies that it has a local truncation error of $O(h^{n+1})$, and therefore a global truncation error of $O(h^n)$. Since we are using the local truncation error form, we apply $p=n+1$ in Eq. (23) for DPM-Solver-n.
>
> The potential issue we anticipated with Runge-Kutta-like solvers is that the performance improvement relative to NFE would become limited with additional NFEs. To address this, we proposed the use of approximation, as detailed in L281-301 of the revised manuscript. The reviewer may have assumed that the truncation error form might change in this context. To explain this, the solution after approximation and extrapolation may not completely eliminate the $O(h^{n+1})$ error term. However, even so, it effectively reduces the original error term, as experimental results (Table 2) show that it leads to a clear performance improvement.
>
> On the other hand, it was difficult for us to infer the reviewer's concerns based solely on this comment. If there are further uncertainties or clarifications needed, we would be happy to address them.
>
> ---
> >Meanwhile I am still concerned potential issues with this algorithm, and whether the local Lipschitz continuity of f and its derivative is too strong.
>
> $f \\in C^2$ is the only assumption we need to meet local Lipschitz continuity of $f$ and $f’$. It is a standard practice to make assumptions about the smoothness of functions when applying numerical techniques, not just in diffusion models but also in solving many physical problems numerically.
> Furthermore, Eq. (3) is an ODE for diffusion models and $f$ can be described as the following from EDM [1]:
> $$
> -t \\cfrac{\\nabla_{x} \\sum_i \\mathcal{N}(x;y_i,t^2 \\mathbf{I})}{\\sum_i \\mathcal{N}(x;y_i,t^2 \\mathbf{I})},
> $$
> where $y_i$ are data points, $t \\in [t_{min}, t_{max}]$ and $t_{min} >0$. It is clearly twice continuously differentiable since $t \\geq t_{min} > 0$. This suggests that the assumption can reasonably extend to other diffusion models as well, and therefore we do not believe this assumption to be overly restrictive. Regarding this content, we have referenced ​​as “the equation in Appendix B.3 of Karras et al. (2022)” in L223-224 of the original submission. In addition, the empirical results of our method also validate the assumption’s appropriateness.
>
> [1] Tero Karras, Miika Aittala, Timo Aila, and Samuli Laine. Elucidating the design space of diffusion-based generative models. In NeurIPS, 2022.

---

> ### Author Response · Authors · 2024-11-30
> **Response (2/2)**
>
> >In regard to the definition of the local truncated error. In Page 4, the authors stated that the local truncation error of .... is given {\hat{x}}^{(1)}{t{i-1}} while defining this as the numerical solution; this will confuse reader. We define error between x_1 and x_0 as error = x_1 - x_0; to me, it is unacceptable to say the error defined as x1 because this obviously confuses readers.
>
> We think describing the use of the expression that $A$ is given by $A+B=C$ as "unacceptable" seems overly dismissive, where $A$ in this case is a local truncation error. More importantly, the equation in question is inherently correct. In our previous response, we also provided a rewritten explanation from the perspective of the definition of local truncation error to further clarify this point. In fact, we would like to add that the way we presented this equation was intentional. As in Eqs (17)-(19), our objective is to obtain numerical ODE solutions and utilize them for extrapolation. Consequently, we arranged the equations with the ODE solution term on the left-hand side to highlight its significance in our approach.
>
>
>
> ---
>
> >The authors still did not pinpoint the reference used for Eqn. (20) and I hope they have revised their paper to make it more easier for the readers. I understand this is re-parameterization of diffusion model, however, I am concerned if there is error in the y form. The reference was not correct in the paper as well.
>
> While we could elaborate further on the re-parameterization of ODE in the main text, this is a well-established concept thoroughly explained in the original DDIM paper, which we cited and directly utilized for this interpretation.
> As for the concern about error in the y form, we respectfully disagree. In our previous response, we provided a detailed explanation showing that the equation is correct, with only a change in notation. Moreover, the same parametrization has been used in other works, such as Eq (6) in GENIE [2].
>
> Finally, regarding the reference, we confirm that we explicitly included a proper citation of the DDIM paper as “DDIM (Song et al.,2021a)” in L243-244 of the revised manuscript. Furthermore, in our previous response, we pinpointed the exact reference within the DDIM paper (Eq (14) and its accompanying descriptions), which we believe should resolve any doubts regarding Eq (20).
>
> [2] Tim Dockhorn, Arash Vahdat, and Karsten Kreis. GENIE: Higher-order denoising diffusion solvers. In NeurIPS, 2022.

---

### Official Review · Reviewer_UczB · 2024-11-02

**Soundness:** 3
**Presentation:** 3
**Contribution:** 3
**Rating:** 6
**Confidence:** 5

**Summary:**

The authors present an enhanced diffusion sampler called RX-DPM, inspired by Richardson extrapolation, which significantly improves the accuracy of existing ODE-based samplers. They create an algorithm for arbitrary discretization specifically designed for DPMs.
• In the paper they outline utilization of the algorithm utilizing Richardson extrapolation for general DPM solvers, accommodating arbitrary time step scheduling, beginning with the derivation of the truncation error formula for the Euler method on a non-uniform grid. They offer implementation details applicable to various diffusion samplers without adding extra function evaluations.
•The experiments with several established baselines reveal that RX-DPM demonstrates robust generalization capabilities and high practicality, irrespective of the ODE structures, architectures, or foundational samplers used.

**Strengths:**

The authors introduce an innovative sampling technique for diffusion models. This method, RX-DPM, appears to surpass previous sampling methods due to its incorporation of Richardson extrapolation, specifically adapted for DPMs. The algorithm effectively reduces local truncation error, resulting in improved sample quality.
The authors demonstrated the effectiveness of RX-DPM through experiments with established baseline models and datasets, as well as by comparing RX-DPM to other sampling methods.

**Weaknesses:**

There is a notable absence of performance comparisons regarding computation time and FLOPS, as the study only evaluates the number of function evaluations (NFE) of the algorithm without contrasting the computational costs of each step of RX-Euler with the standard Euler method for DPMs.
Furthermore, most of the experiments involving NFE variation focus on low to moderate-resolution images, not exceeding 256 by 256 pixels, while leading diffusion models are capable of generating high-resolution images, such as 512 by 512 pixels or more. Consequently, the study lacks a thorough quality comparison of the methods across high-resolution datasets like AFHQv2 and FFHQ, particularly for varying NFEs beyond 10 or 11

**Questions:**

What is the difference of the proposed method with alternative methods in the literature in terms of sampling time?
Why extrapolation does not require additional NFE? In which scenario us the method is applicable for SDE?

---

> ### Author Response · Authors · 2024-11-17
> **Response**
>
> Thank you for taking the time and effort to review our work. We have addressed the raised concerns and questions and updated the manuscript to improve clarity. Please let us know if further clarification is needed.
> ***
> > W1. There is a notable absence of performance comparisons regarding computation time and FLOPS, as the study only evaluates the number of function evaluations (NFE) of the algorithm without contrasting the computational costs of each step of RX-Euler with the standard Euler method for DPMs.
>
> In the table below, we compare the computation times of the Euler method and RX-Euler using EDM backbone. The average runtime per batch is measured for 10-step sampling with a batch size of 128.
> The additional operations introduced by our method, which consist of linear combinations of precomputed values, result in negligible computational overhead compared to the time required for the network forward pass.
> Furthermore, as the model size increases, the relative overhead diminishes (e.g., only 0.11\% increase for ImageNet class-conditional sampling. Therefore, comparing sampling times in terms of NFE is sufficient. Meanwhile, recognizing its value to the paper, we have included this content in Appendix E of the revised manuscript.
>
> __Comparison of average runtime / batch (seconds)__
> | |CIFAR-10 cond. (32x32)|FFHQ (64x64)|ImageNet cond. (64x64)|
> |--|:--:|:--:|:--:|
> |Euler |1.737 $\\pm$ 0.028|3.895 $\\pm$ 0.023|6.436 $\\pm$ 0.033|
> |RX-Euler |1.743 $\\pm$ 0.031|3.903 $\\pm$ 0.025|6.443 $\\pm$ 0.039|
> ***
> > W2. Furthermore, most of the experiments involving NFE variation focus on low to moderate-resolution images, not exceeding 256 by 256 pixels, while leading diffusion models are capable of generating high-resolution images, such as 512 by 512 pixels or more. Consequently, the study lacks a thorough quality comparison of the methods across high-resolution datasets like AFHQv2 and FFHQ, particularly for varying NFEs beyond 10 or 11.
>
> In Section 5.4 of our original submission, we conducted text-to-image experiments using Stable Diffusion to generate 512x512-sized images, demonstrating the validity of RX-DPM in large and high-resolution datasets. We believe this provides sufficient evidence of our method's scalability. Also, experiments on high resolutions in this field are typically demonstrated using Stable Diffusion and we believe it is unlikely to exhibit a different trend at larger resolutions.
>
> ***
> > Q1. What is the difference of the proposed method with alternative methods in the literature in terms of sampling time? Why extrapolation does not require additional NFE?
>
> Newly introduced operations by RX-DPM consist of two parts: calculating 1-step estimation for k-step interval (line 11 of Algorithm 1), and performing extrapolation (line 12 of Algorithm 1). For calculating 1-step estimation, we have explained that the Euler method does not require extra NFEs on L251-254. Moreover we have provided examples to calculate 1-step estimation without requiring model evaluation for Runge-Kutta and Adam-Bashforth methods in Section 4.3. Note that the most ODE solvers for diffusion models fall on these categories. Moreover, performing extrapolation is a linear combination of k-step estimation and 1-step estimation, which are already calculated. Therefore, RX-DPM does not incorporate any network evaluation and is achieved by calculating a few linear combinations of saved values. Furthermore, as demonstrated in our response to W1 regarding computational times, performing linear combinations is negligible compared to the network evaluation.
> ***
> >Q2. In which scenario is the method applicable for SDE?
>
> To incorporate our method with an SDE solver,  we interpret the SDE solver as a combination of a deterministic sampling component and a stochasticity component and we utilize the deterministic sampling part as an ODE solver. Specifically, within each $k$-step interval (the basic unit where extrapolation occurs), we execute the RX-DPM algorithm using deterministic sampling taken from the SDE solver and add stochasticity term afterward. (For a better understanding of our implementation, we provide the diagrams of ODE and the proposed method in Appendix C of the revised manuscript.) Consequently, we leverage the effect of accurate mean prediction of RX-DPM while partially incorporating the mean correction effect that stochasticity provides. This effect can vary depending on the characteristics of the trained models and datasets. When the mean correction from stochasticity is large and significant, there may be some offsetting effect with our method. However, when the mean correction effect from stochasticity is relatively small, our method harmonizes effectively with SDE sampling, producing excellent results. As shown in Table 4 and described in L528 of our original submission, particularly when the NFE is very low, applying our method can significantly enhance the performance.

---

> ### Author Response · Authors · 2024-11-25
> **Additional qualitative results**
>
> In response to the reviewer's comments regarding quality comparison, we have added more qualitative results across a wider range of NFEs in Appendix F of the revised manuscript. Specifically, we included results for Stable Diffusion (512x512 resolution) under more restricted NFE settings. Overall, the results demonstrate that the blurry and desaturated appearance observed in DDIM at low NFEs becomes noticeably sharper and more vivid when our method is applied. We hope these additional results address concerns regarding the absence of high-resolution quality comparisons. If you have any further questions or need additional clarification, we would be glad to address them.

---

### Official Review · Reviewer_6Qbz · 2024-11-03

**Soundness:** 2
**Presentation:** 3
**Contribution:** 3
**Rating:** 6
**Confidence:** 3

**Summary:**

Inspired by the Richardson extrapolation, the author propose an enhanced ODE-based sampling method called RX-DPM for DPMs to reduce numerical errors and improve convergence rates. The author outline the algorithmic development process for the most simpliﬁed problem and explore an extension to a general DPM solver. The experiments show that RX-DPM can reduce the truncation error and achieve better sample qualities.

**Strengths:**

1. propose an diffusion sampling method called RX-DPM, inspiring by richardson extrapolation. The method can achieve accurate estimation of numerical solutions without additional computational overhead.Besides, the method is applicable to arbitrary discretizations of time steps.

2. The experiments show that RX-DPM can reduce the truncation error and achieve better sample qualities. RX-DPM shows strong generalization and practical effectiveness, performing well across various ODE designs, architectures, and base samplers.

**Weaknesses:**

1. The value of $p$ is not provided, and the experiment does not show how the sample quality changes as order $p$ increases.

**Questions:**

1. How to apply propose method into the SDE sampler?

---

> ### Author Response · Authors · 2024-11-17
> **Response**
>
> Thank you for taking the time to review our work. We have addressed the questions and concerns raised in detail and have updated the manuscript to enhance clarity and understanding. ​​If anything remains unclear, please let us know, and we will be happy to provide further clarification.
> ***
> >1. The value of $p$ is not provided, and the experiment does not show how the sample quality changes as order $p$ increases.
>
> $p$ represents the order of the leading term of the local truncation error. Thus, it is a fixed value determined by the choice of the baseline ODE solver. We show the value of $p$ for solvers used in the paper in the following table.
>
> | Solver | Euler / DDIM | DPM-solver-2 | DPM-solver-3 | S-PNDM | F-PNDM |
> |--------|:------------:|:------------:|:------------:|:------:|:------:|
> | p      |       2      |       3      |       4      |    3   |    5   |
>
> To avoid confusion, this has been specified in Section 5.5 as follows in L457 and L463 of the original submission, respectively:
>
> - Note that single-step DPM-solver-$n$ can be considered as an $n^{\\text{th}}$ Runge-Kutta-like solver; therefore we apply RX-DPM with $p=n+1$ for DPM-solver-$n$.
>
> - S-PNDM and F-PNDM utilize the linear multistep methods (i.e, $2^{\\text{nd}}$ and $4^{\\text{th}}$ Adam-Bashforth method, respectively) except for the first few steps; thus we apply RX-DPM with $p=3$ and $p=5$ for S/F-PNDM, respectively.
>
> ***
> >2. How to apply propose method into the SDE sampler?
>
> The proposed approach is essentially compatible with ODE solvers. To incorporate our method with SDE solver,  we interpret an SDE solver as a combination of a deterministic sampling component and a stochasticity component and we utilize the deterministic sampling part as an ODE solver. Specifically, within each $k$-step interval (the basic unit where extrapolation occurs), we execute the RX-DPM algorithm using deterministic sampling taken from the SDE solver and add stochasticity term after the extrapolation.
> For example, in Section 5.6, NPR-DDIM and SN-DDIM methods applied in our experiments propose to learn optimal variances that apply to the deterministic version of DDIM. Therefore, after applying RX-DDIM every $k=2$ steps, we added the optimal variance suggested by NPR/SN-DDIM to perform sampling as described in L482-485 of our original submission. For better understanding of our implementation, we have included the diagrams of ODE and the proposed method in Appendix C of the revised manuscript.

---

> > ### Comment · Reviewer_6Qbz · 2024-11-26
> >
> > I thank the authors to solve my concerns. I have no doubt on the equation in this paper. Besides, the added diagrams make the RX-DPM more clear. Overall, I choose to maintain my current score.

---

> > > ### Author Response · Authors · 2024-11-26
> > >
> > > We thank the reviewer for the response and are pleased that the concerns have been resolved. In particular, we are grateful for the careful review and feedback on the equations.

---

### Official Review · Reviewer_Fth1 · 2024-11-04

**Soundness:** 2
**Presentation:** 3
**Contribution:** 3
**Rating:** 6
**Confidence:** 3

**Summary:**

The work considers the problem of accelerating the inference of diffusion probabilistic models (DPM). Different from the distillation methods, the work adopts the ODE perspective of DPM and aims to propose an accelerated solver for solving the ODE associated with DPMs. The particular technique the work leverages is the Richardson extrapolation (RX), which is proven to achieve lower numerical errors given the same amount of iterations. Analogous to the RX in Euler, the authors derive the ODE solver for DPMs (eq. 22-23). Note that a rigorous mathematical justification of eq 22-23 is not presented, but a simple derivation of the truncation error in terms of total iteration number N is given (assuming eq. 22-23 to be true).

Two different experiments are conducted to demonstrate the performance of RX-DPMs. The first experiment focus on the RX-Euler case, which can be derived rigorously. The experiment clearly shows that RX-Euler leads to faster convergence. The second experiments focuses on RX-DPMs. The results demonstrate faster convergence of RX-DPMs than other DPMs, which, however, is not as significant as RX-Euler over Euler.

**Strengths:**

1. The paper is generally easy to follow.
2. The introduction of Richardson extrapolation is interesting.

**Weaknesses:**

1. The limitations of the proposed method is not well discussed.
2. The justification of equation 22 needs improvement on its clarity and rigor.
3. The second weakness leads to a disconnection in the logic flow from RX-Euler to RX-DPMs, which reduces the significance of the second contribution considered by the authors, that is, the proposal of an RX method on a non-uniform grid.

**Questions:**

---- Methodology ----
1. The authors are advised to define what truncation error means in the first place, as readers may have different interpretations of the error.
2. As pointed out in the weakness, the authors are advised to provide a more clear and rigorous justification of eq. 22-23.
3. Some symbols are used before proper definition, such as $\lambda$.

---- Experiment -----
1. [Fig.2] Please show the dimensionality of the images from CIFAR-10 and FFHQ.
2. [Fig.3] The authors are advised to explain why not all algorithms start at NFE=0.
3. A paragraph properly discussing the limitation of the proposed method is recommended. Besides RX may not be extended to SDE, another clear limitation of the method is the lack of rigorous mathematical justification.
4. The current experiment does not include any other fast DPM methods. To help properly evaluate the performance, the authors are advised to compare RX-DPMs with either a distillation-model-type or consistency-model-type method.

---

> ### Author Response · Authors · 2024-11-17
> **Response to Weaknesses and Questions about Methodology**
>
> Thank you for taking the time to review our work. We have carefully addressed the questions and concerns raised, and have made updates to the manuscript to improve clarity and understanding. If any points remain unclear, please let us know, and we will be happy to provide further clarification.
> ***
> >W1. The limitations of the proposed method is not well discussed.
>
> While we have addressed the limitations throughout the main text, we agree that discussing them separately is a valuable suggestion. Accordingly, we will consolidate these discussions and include them in the revised manuscript as follows:
>
> Our method is primarily designed for an ODE solver. To integrate it with an SDE, we partially incorporate the stochastic component of the SDE solver, as described in Section 5.6. Consequently, its effectiveness may be reduced in some cases, as the full impact of stochasticity is not captured. However, in scenarios with highly limited NFE, which are of greater interest to us, the combined effect of RX-DPM and stochastic sampling has been shown to be beneficial. Developing methods that perform better in more general cases is left as future work. Additionally, in extending the RX-Euler algorithm to higher-order solvers in Section 4.3, there is room for improvement, as we impose assumptions on linearly accumulated errors. Relaxing these assumptions or deriving more accurate formulations could further enhance performance.
> ***
>
> >W2. The justification of equation 22 needs improvement on its clarity and rigor.
>
> >W3. The second weakness leads to a disconnection in the logic flow from RX-Euler to RX-DPMs, which reduces the significance of the second contribution considered by the authors, that is, the proposal of an RX method on a non-uniform grid.
>
> To address potential misunderstandings, we have clarified the explanation leading to Eq.(22) and revised the manuscript accordingly, recognizing that the initial wording could be misleading. Unlike the simple case of the Euler method, for arbitrary high-order solvers, it is not feasible to rigorously derive the intermediate terms with $h$. To address this, we propose an algorithm that uses Eq.(22), which is a natural extension of Eq.(18) and follows from the assumption of linear error accumulation, as an approximate alternative. Although we can not provide full mathematical rigor, our approach is reasonable in that using this equation is consistent with the assumptions inherent in the standard Richardson extrapolation method as we mentioned in L271-272 and Appendix A of the original submission. Furthermore, our approach provides a simple and practical algorithm that is generalizable and easy to implement. To eliminate any potential misunderstandings, we revised the expression in L262 of the original submission as the linear error accumulations are assumed in Eq.(22) and elaborated more on this in Appendix B of the revised manuscript.
>
> ***
> ---- Methodology ----
>
> >Q1. The authors are advised to define what truncation error means in the first place, as readers may have different interpretations of the error.
>
> In response to the reviewer’s suggestion, we have included a clear definition of truncation error as the error caused by one-step approximation, ensuring that its meaning is unambiguous for all readers in L187 of the original submission.
> ***
> >Q2. As pointed out in the weakness, the authors are advised to provide a more clear and rigorous justification of eq. 22-23.
>
> We believe that we have addressed the explanation related to Eq.(22) in response to [W1] above. Eq.(23) is derived by solving the linear system formed by Eq.(21) and (22) by eliminating the $h^p$ term and leading to the resulting Eq.(23).
> ***
> >Q3. Some symbols are used before proper definition, such as $\lambda$.
>
> $\\lambda$ is properly defined before referencing in L193 in our original submission. Please note that this notation is used consistently throughout the paper; therefore, it was omitted in subsequent sections.

---

> ### Author Response · Authors · 2024-11-17
> **Response to Questions about Experiment**
>
> ---- Experiment -----
> >Q1.  [Fig.2] Please show the dimensionality of the images from CIFAR-10 and FFHQ.
>
> In response to the reviewer’s comment, we have included the dimensionality of the images in Figure 2 in the revised version of our manuscript.
> ***
> >Q2. [Fig.3] The authors are advised to explain why not all algorithms start at NFE=0.
>
> RX-DPM requires at least two steps, as it utilizes two different ODE solutions. Additionally, since we rely on pretrained models and do not perform any training, the minimum number of steps are required for the pretrained model to function adequately. While we could demonstrate improvements with smaller NFEs, we believe it is more meaningful to compare the performances within the range where decent quality is achieved. Please note that this is also common in other papers that propose improved solvers.
> ***
> >Q3. A paragraph properly discussing the limitation of the proposed method is recommended. Besides RX may not be extended to SDE, another clear limitation of the method is the lack of rigorous mathematical justification.
>
> Please refer to our response to W1.
> ***
> >Q4. The current experiment does not include any other fast DPM methods. To help properly evaluate the performance, the authors are advised to compare RX-DPMs with either a distillation-model-type or consistency-model-type method.
>
> We compare the results with additional methods in the table below on ImageNet 64x64. Because each model utilizes different architectures and it is practically difficult to implement and reproduce all algorithms, we collected available outcomes from various references. Our application demonstrated that our approach outperforms other diffusion acceleration methods using a shared backbone. We would like to also note that our approach can be considered orthogonal to distillation-based methods. While distillation-based methods require model training supervised by the pre-trained model, our algorithm is an agnostic sampling algorithm applied to arbitrary ODE-based sampling methods. Thus, we focus on generality across diverse well-known solver-based sampling acceleration methods including DPM-Solver and PNDM. Additionally, we believe our method can serve as an improved teacher since many distillation-based methods utilize ODE samplers during training (eg., consistency models achieve better results when using Heun’s method than Euler for training), thus we expect that combining our approach with distillation-based methods can potentially yield better results.
>
> __Imagenet 64x64 cond.__
>
> | Method            | NFE |  FID  |    Backbone    |
> |-------------------|:---:|:-----:|:--------------:|
> | ADM [1]           | 250 |  2.07 |       ADM      |
> | Analytic-DDIM [2] | 100 | 17.73 |   iDDPM [11]   |
> | GENIE [3]         |  10 |  7.41 | Score-SDE [12] |
> | S-PNDM [4]        |  10 | 11.20 |      iDDPM     |
> | EDM [5]           |  79 |  2.44 |       EDM      |
> | LA-DPM [6]        |  10 |  9.89 |       EDM      |
> | RX-Euler          |  10 |  6.95 |       EDM      |
> | DPM-Solver-3 [7]  |  12 |  5.02 |      iDDPM     |
> | DEIS [8]          |  12 |  3.99 |      iDDPM     |
> | RX-DPM-Solver-3   |  12 |  3.90 |      iDDPM     |
> | PD [9]            |  1  | 15.39 |      iDDPM     |
> | CD [10]           |  1  |  6.2  |       ADM      |
>
> [1] Dhariwal, P. and Nichol, A., Diffusion models beat GANs on image synthesis. In NeurIPS, 2021.
>
> [2] Bao, F. et al., Analytic-DPM: an Analytic Estimate of the Optimal Reverse Variance in Diffusion Probabilistic Models. In ICLR 2022
>
> [3] Dockhorn, T., Vahdat, A., and Kreis, K., GENIE: Higher-Order Denoising Diffusion Solvers. In NeurIPS, 2022.
>
> [4] Liu, L., Ren, Y., Lin, Z., and Zhao, Z., Pseudo numerical methods for diffusion models on manifolds. In ICLR, 2022.
>
> [5] Karras, T. et al., Elucidating the design space of diffusion-based generative models. In NeurIPS, 2022.
>
> [6] Zhang, G., Kenta, N., and Kleijn, W. B., Lookahead diffusion probabilistic models for refining mean estimation, 2023.
>
> [7] Lu, C. et al., DPM-Solver: A fast ODE solver for diffusion probabilistic model sampling in around 10 steps. In NeurIPS, 2022.
>
> [8] Zhang, Q., and Chen, Y., Fast sampling of diffusion models with exponential integrator, 2023.
>
> [9] Salimans, T. and Ho, J., Progressive distillation for fast sampling of diffusion models. In ICLR, 2022
>
> [10] Song, Y., et al., Consistency models. In ICML, 2023.
>
> [11] Nichol, A. and Dhariwal, P., Improved Denoising Diffusion Probabilistic Models. In ICML, 2021.
>
> [12] Song, Y., Sohl-Dickstein, J., Kingma, D. P., Kumar, A., Er- mon, S., and Poole, B. Score-based generative modeling through stochastic differential equations. In ICLR, 2021.

---

> > ### Comment · Reviewer_Fth1 · 2024-11-25
> >
> > I thank the authors for revising the manuscript and performing extra experiments to clarify my concerns. I am still concerned with the lack of mathematical rigor in RX-DPM, which is also shared by Reviewer xBMM. Furthermore, I acknowledge that different accelerated diffusion models are based on different checkpoints, but the performance of the proposed methods remains unambiguous without having a proper comparison (e.g. same checkpoint, control NFE and compare FID, and control FID compare NFE). Given these weakness, I lean to keep my score as it is.

---

> > > ### Author Response · Authors · 2024-11-25
> > > **Response**
> > >
> > > >I thank the authors for revising the manuscript and performing extra experiments to clarify my concerns. I am still concerned with the lack of mathematical rigor in RX-DPM, which is also shared by Reviewer xBMM.
> > >
> > > Thank you for your response. However, we respectfully disagree with the concerns raised regarding the rigor of RX-DPM. All derivations, except for Eq. (22), are either proved in the paper, or supported by references. We have clarified that the equations questioned by Reviewer xBMM are correct. Specifically, Eq. (10) is local truncation error analysis of the Euler method, and it is well-described in many numerical analysis literature. We feel underestimating mathematical rigor based on Eq. (10) is inappropriate. To support this, we kindly request you to double-check Eq. (10):
> > >
> > > $$
> > > \\hat x_{t_{i-1}}^{(1)}
> > > = x_{t_i} -\\lambda_1 h f(x_{t_{i}};t_{i})
> > > = x_{t_{i-1}}^{*} - \\frac{1}{2} x_{t_{i}}'' \\lambda_1^2 h^2 + O(h^3).
> > > $$
> > >
> > > On the other hand, we have clarified in the revised manuscript that Eq. (22) is given by an assumption. In particular, when the time steps are even, Eq. (22) is equivalent to the assumption of conventional Richardson extrapolation, as detailed in Appendix B of revised manuscript, making our approach reasonable. Developing algorithms under reasonable assumption is a common practice, and the results in Sec. 5.5 also show the validity of the suggested algorithm.
> > >
> > > ---
> > > >Furthermore, I acknowledge that different accelerated diffusion models are based on different checkpoints, but the performance of the proposed methods remains unambiguous without having a proper comparison (e.g. same checkpoint, control NFE and compare FID, and control FID compare NFE).
> > >
> > > In the table we provided, except for distillation-based methods, models that share the same backbone use identical checkpoints. It is explicitly stated in the respective papers or official codebases that each model utilized the official implementation of the backbone model along with the provided checkpoints. As noted in Sec. 5.1, our method followed the same approach to ensure a fair comparison. The table we provided demonstrates consistent performance improvements over other acceleration methods within models sharing the same backbone and checkpoint.
> > >
> > > The distillation-based methods (PD[9], CD[10]) suggested by the reviewer for comparison require model training, making direct comparisons using the same checkpoint impossible. Therefore, we relied on the reported results and publicly available values provided in their respective papers.

---

> > > ### Author Response · Authors · 2024-11-29
> > >
> > > Dear reviewer Fth1,
> > >
> > > We kindly request clarification regarding any remaining unresolved concerns.
> > > Meanwhile, reviewer 6Qbz has expressed that there are no problems with the equations in the paper.
> > > We would appreciate your feedback on any aspects you have doubted about.
> > >
> > > Best regards,
> > >
> > > authors

---

### Author Response · Authors · 2024-11-18
**Manuscript revisions**

We appreciate all the reviewers for taking the time and effort to provide valuable feedback. We have revised the manuscript to reflect the reviewers' suggestions and comments. The major changes include:

- A separate section for limitations in Appendix A.
- Diagrams of the baseline and proposed algorithms in Appendix C.
- A comparison of computation times in Appendix D.
- Restructured content and additional explanations in Section 4.3 for enhanced clarity and readability.

Additionally, any revised or newly added text for improved clarity has been highlighted in blue.
We are also happy to address any further questions or concerns that may arise.

---

### Author Response · Authors · 2024-11-19

We summarize the key points of our responses as follows:

- A more detailed explanation of how the proposed method is applied to SDE solvers, along with a discussion of limitations, addressing feedback from all reviewers.
- A detailed explanation and justification of Eq.(22) in response to reviewer Fth1.
- An in-depth discussion of computational complexity and a comparison of actual computation times, addressing comments from reviewers UczB and xBMM.
- A step-by-step derivation of Eq.(15) in response to reviewer xBMM.

In addition to these points, we have incorporated suggestions regarding the presentation and clarified any potential misunderstandings raised in the reviews. These updates have been reflected in the revised manuscript, which has been uploaded.

We believe we have addressed every concern and question raised. However, if there are any remaining doubts or unresolved issues, we would greatly appreciate your feedback and would be happy to provide further clarification.

---

### Author Response · Authors · 2024-11-24

Dear reviewers,

As we approach the end of the discussion period, we would like to follow up on the points we have addressed in our rebuttal. If there are any remaining questions or unresolved issues, we would be happy to provide further clarification to ensure your concerns are fully addressed. We kindly ask you to review our latest responses and revised manuscript, which aim to address each of the points you raised. Your feedback is invaluable, and we sincerely appreciate your time and effort throughout this process.

Best regards,

authors

---

### Meta-Review · Area_Chair_fHAi · 2024-12-16

**Metareview:**

This paper introduces RX-DPM, a sampling method for diffusion probabilistic models that significantly enhances the efficiency and accuracy of the sampling process. By leveraging Richardson extrapolation techniques adapted for non-uniform time steps, RX-DPM uses multiple ODE solutions to improve the quality of generated samples without increasing the number of function evaluations. The method also provides explicit error estimates, demonstrating faster convergence rates and improved numerical accuracy. Through convincing numerical experiments, the method is validated across various settings.

This work is particularly important and timely since it addresses the high computational costs associated with DPM sampling, a key limitation in scaling such models. The proposed method is straightforward to implement and does not require retraining or fine-tuning of the original model, making it complementary to emerging lines of research such as consistency models (ie. methods that do require retraining of the models). Furthermore, the paper’s approach to adapting well-understood Richardson extrapolation techniques for arbitrary time step schedules and its integration of error estimation provide a solid theoretical foundation. The numerical results are convincing, demonstrating the practicality and robustness of the method across different setups.

The panel of reviewers recommends acceptance of this paper. It makes a meaningful contribution to the field by providing a simple yet effective solution to a fundamental challenge in DPMs. The combination of theoretical rigor, practical and ease of implementation, and strong experimental validation makes RX-DPM a valuable addition to the community.

**Additional Comments On Reviewer Discussion:**

During the rebuttal period, the authors provided extensive clarifications on the theoretical foundations of their proposed method, RX-DPM, addressing concerns raised by the reviewers. They expanded the comparisons with competing methods, offering additional experimental evidence that strengthened the empirical support for their claims. Notably, there was a detailed exchange with Reviewer xBMM regarding the theoretical soundness of the approach. After carefully reviewing the authors’ responses and the ensuing discussion, I ultimately concur with the authors’ position that the method is theoretically sound, to the best of my understanding and evaluation. The rebuttal process was thorough and significantly enhanced the clarity and robustness of the submission.

---

### Decision · Program_Chairs · 2025-01-22

Accept (Poster)